# RBMX2 links *Mycobacterium bovis* infection to epithelial–mesenchymal transition and lung cancer progression

Chao Wang[1,2,3†], Yongchong Peng[1,2†], Hongxin Yang[4,5†], Yanzhu Jiang[1,2], Abdul Karim Khalid[1], Kailun Zhang[1,2], Shengsong Xie[1], Luiz Bermudez[6], Yong Yang[7], Lei Zhang[1,2], Huanchun Chen[1], Aizhen Guo[1,2,8]*, Yingyu Chen[1,2,8]*

[1]The National Key Laboratory of Agricultural Microbiology, College of Veterinary Medicine, Huazhong Agricultural University, Wuhan, China; [2]National Animal Tuberculosis Para-Reference Laboratory (Wuhan) of Ministry of Agriculture and Rural Affairs, Huazhong Agricultural University, Wuhan, China; [3]Center for Infectious Disease Research, School of Medicine, Westlake University, Hangzhou, China; [4]Oncology Collaborative Innovation Center, Inner Mongolia Medical University, Hohhot, China; [5]Department of Cell Biology, Inner Mongolia Medical University, Hohhot, China; [6]Department of Biomedical Sciences, College of Veterinary Medicine, Oregon State University, Corvallis, United States; [7]Department of Surgical Oncology, People's Hospital of Inner Mongolia Autonomous Region, Hohhot, China; [8]Hubei Hongshan Laboratory, Huazhong Agricultural University, Wuhan, China

*For correspondence:
aizhen@mail.hzau.edu.cn (AG);
chenyingyu@mail.hzau.edu.cn (YC)

†These authors contributed equally to this work

Competing interest: The authors declare that no competing interests exist.

## eLife Assessment

The identification of RBMX2 as a novel regulator linking mycobacterial infection to Epithelial-Mesenchymal Transition and cancer progression are **fundamental** findings that advance our understanding of a major research question about the link between infectious and non-infectious diseases, microbiology and oncology. It does so by introducing RBMX2 as a novel host factor, a potential therapeutic target and biomarker for both TB and lung cancer. The evidence provided is **convincing** because it is appropriate and the validated multi-omics methodologies used are in line with the current state of the art. This study will be of interest to scientists working in the fields of drug discovery, microbiology and oncology.

**Abstract** Tuberculosis (TB) is a complex disease caused by the interaction of pathogen, host, and environmental factors. In 2022, TB affected 10.6 million people and caused 1.3 million deaths globally. In high-burden zoonotic TB regions, *Mycobacterium bovis* accounts for ~10% of human TB cases. The immune evasion and latency of *Mycobacterium tuberculosis* hinder understanding of host responses. Here, we identify RNA-binding motif protein X-linked 2 (RBMX2) as a novel host factor facilitating *M. bovis* infection. RBMX2 expression is significantly upregulated in multiple cell types, including EBL, BoMac, bovine alveolar primary cells, and human A549 cells. Multi-omics analyses, cell adhesion assays, and ChIP-PCR demonstrate that RBMX2 suppresses cell adhesion and tight junctions while enhancing *M. bovis* adhesion and invasion via p65 signaling. Integrated transcriptomic, proteomic, and metabolomic data reveal that RBMX2 regulates epithelial–mesenchymal transition (EMT), a process linked to cancer progression. TIMER2.0 analysis shows elevated RBMX2 expression in lung adenocarcinoma and lung squamous cell carcinoma tissues, validated by immunofluorescence. Using an *M. bovis*-induced BoMac-EBL EMT model and H1299 cells, we show that RBMX2 promotes EMT through p65/MMP-9 pathway activation. Collectively, RBMX2 is a novel host

factor that enhances *M. bovis* infection and drives infection-induced EMT. These findings provide new insight into TB pathogenesis and highlight RBMX2 as a potential target for TB vaccine and therapeutic development.

## Introduction

Tuberculosis (TB) remains a major global health threat, with approximately 10.6 million new cases and 1.3 million deaths reported worldwide in 2022 (*WHO, 2023*). *Mycobacterium tuberculosis* and its zoonotic variant, *Mycobacterium bovis*, are responsible for a significant proportion of TB cases, particularly in regions with a high burden of zoonotic diseases, where *M. bovis* accounts for up to 10% of human TB infections. The immune evasion strategies of *M. tb* and its variants complicate our understanding of host immune responses and disease progression.

Recent studies have increasingly linked chronic infections, including TB, to cancer development (*Preda et al., 2023*; *Cabrera-Sanchez et al., 2022*). The persistent inflammatory environment induced by chronic infection can drive cellular transformations, such as epithelial–mesenchymal transition (EMT), which is closely associated with cancer initiation and metastasis (*Suarez-Carmona et al., 2017*). However, the specific molecular mechanisms and host factors involved in this infection–cancer axis remain largely uncharacterized.

In this context, RNA-binding motif protein X-linked 2 (RBMX2) has emerged as a potential host factor involved in both TB infection and cancer progression. Our previous studies demonstrated that RBMX2 regulates alternative splicing of APAF-1 introns, thereby modulating apoptosis following *M. bovis* infection (*Wang et al., 2024*). Furthermore, accumulating evidence suggests that its homolog, RBMX, plays dual and context-dependent roles in tumorigenesis, acting as either an oncogene or a tumor suppressor. For example, RBMX is overexpressed in hepatocellular carcinoma (*Song et al., 2020*) and T-cell lymphomas (*Schümann et al., 2021*), whereas its expression is reduced in pancreatic ductal adenocarcinoma, indicating a complex and tissue-specific regulatory function (*Alors-Pérez et al., 2024*).

Despite these findings, the role of RBMX2 in *M. bovis* infection and its potential involvement in EMT regulation remains largely unexplored. Based on our preliminary data, we hypothesize that RBMX2 promotes *M. bovis* adhesion and invasion by modulating host cell properties and further facilitates EMT through activation of the p65/MMP-9 signaling pathway. This study aims to elucidate the molecular mechanisms by which RBMX2 contributes to *M. bovis* infection and its potential role in infection-associated EMT, thereby providing novel insights into the intersection of TB and lung cancer. These findings may highlight RBMX2 as a promising therapeutic target for TB and TB-related malignancies.

## Results

### Elevated expression of RBMX2 in *M. bovis*-infected cells

bovis was used to infect the EBL cell library, which was constructed using CRISPR/Cas9 technology in our laboratory, at a multiplicity of infection (MOI) of 100:1. We evaluated cell survival and performed high-throughput sequencing on the viable cells to identify key host factors potentially involved in anti-*M. bovis* defense. Notably, RBMX2 knockout cells exhibited significant resistance to infection.

Further analysis using AlphaFold Multimer revealed that amino acid residues 56–134 of RBMX2 contain an RNA recognition motif (*Figure 1—figure supplement 1A*). Additionally, we identified 611 RBM family sequences at the bovine protein exon level and classified them into subfamilies based on motif similarity. RBMX2 was predominantly assigned to subfamily VII, which includes proteins such as EIF3G, RBM14, RBM42, RBMX44, RBM17, PUF60, SART3, and RBM25 (*Figure 1—figure supplement 1B*). Comparative analysis of amino acid sequences indicated a high degree of conservation of RBMX2 across species (*Supplementary file 1*).

To further validate RBMX2 expression in EBL cells following infection with *M. bovis* and *M. bovis* Bacillus Calmette–Guérin (BCG), RNA samples were collected at 24, 48, and 72 hr post-infection (hpi). Real-time quantitative polymerase chain reaction (RT-qPCR) results showed that RBMX2 expression was significantly upregulated after infection with both *M. bovis* and *M. bovis* BCG (*Figure 1A, B*). Moreover, increased RBMX2 expression was observed in bovine monocyte-derived macrophages (BoMac cells), bovine alveolar type II primary cells, and human lung epithelial cells (A549 cells)

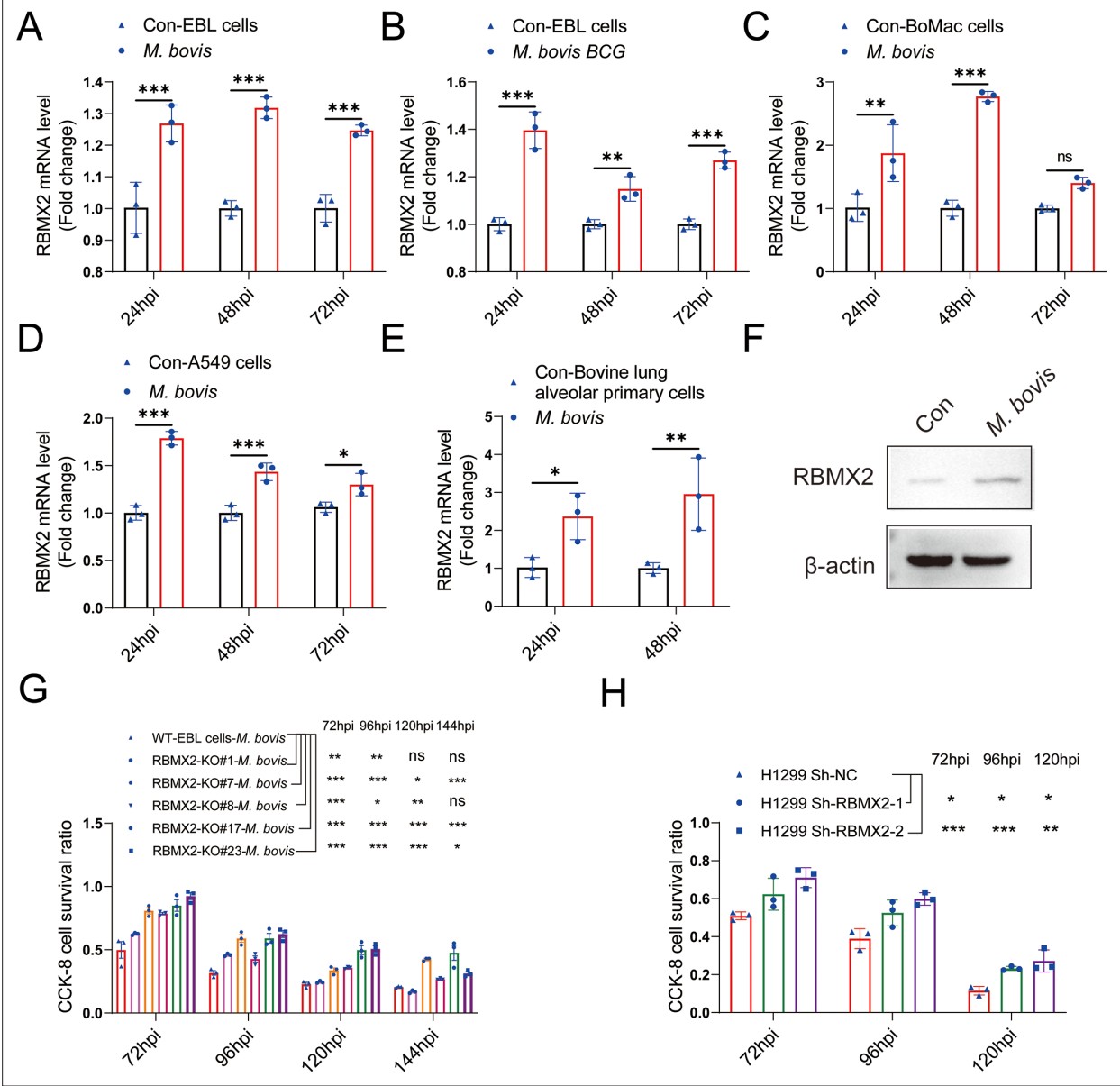

**Figure 1.** Expression of *RBMX2* after infection, and *RBMX2* did not affect cell proliferation but inhibited cell survival during *M. bovis* infection. The expression of *RBMX2* in EBL cells infected by (**A**) *M. bovis* and (**B**) *M. bovis* Bacillus Calmette–Guérin (BCG) was analyzed by real-time quantitative polymerase chain reaction (RT-qPCR). Data were represented by fold expression relative to uninfected cells. (**C–E**) The expression of *RBMX2* mRNA in (**E**) BoMac cells, (**F**) A549 cells, and (**G**) bovine lung alveolar primary cells infected by *M. bovis* was analyzed via RT-qPCR. Data were represented by fold expression relative to uninfected cells. (**F**) The expression of *RBMX2* in EBL cells infected by *M. bovis was* analyzed by WB. Data were represented by fold expression relative to uninfected cells. (**G**) Detection of the ability of different *RBMX2* knockout site monoclonal EBL cells against *M. bovis* infection by CCK-8 assay. Data were represented by the absorbance value relative to WT EBL cells after *M. bovis* infection. (**H**) Detection of the ability of *RBMX2* slicing H1299 cells against *M. bovis* infection by CCK-8 assay. Data were represented by the absorbance value relative to Sh-NC H1299 cells after *M. bovis* infection. One- and two-way ANOVA were used to determine the statistical significance of differences between different groups. Ns presents no significance; *p < 0.05, **p < 0.01, and ***p < 0.001 indicate statistically significant differences. Data were representative of at least three independent experiments.

The online version of this article includes the following source data and figure supplement(s) for figure 1:

**Source data 1.** Original western blots for panel F, indicating the relevant bands.

**Source data 2.** Original files for western blot analysis displayed in panel F.

**Figure supplement 1.** Protein structure analysis, subfamily localization.

**Figure supplement 2.** The influence of *RBMX2* in cell cycle, cell morphology, cell proliferation, and resistance to *M. bovis* infection.

following *M. bovis* infection (*Figure 1C–E*). Furthermore, we used WB to verify the expression levels of RBMX2 protein in EBL cells after *M. bovis* infection. The results showed that *M. bovis* infection upregulates the expression of RBMX2 protein in EBL cells (*Figure 1F*).

## *RBMX2* did not affect cell cycle but inhibited epithelial cell survival during *M. bovis* infection

In generating *RBMX2* monoclonal knockout (KO) cells, monoclonal cells were selected through limited dilution in 96-well plates. Sanger sequencing was conducted to identify different monoclonal cells with distinct knockout sites (*Figure 1—figure supplement 2A*). We validated the effect of RBMX2 knockout on the cell cycle using flow cytometry, and the results indicated that RBMX2 does not affect the cell cycle (*Figure 1—figure supplement 2B*).

To investigate the potential role of *RBMX2* in resistance to *M. bovis* infection, we assessed the survival rate of EBL cells and H1299 cells at different hours post-infection using CCK-8 assay. The findings revealed an enhanced survival rate in both *RBMX2* knockout monoclonal EBL cells following *M. bovis* infection (*Figure 1G*). Furthermore, RBMX2-silenced H1299 cells exhibited a higher survival rate compared to H1299 ShNc cells after *M. bovis* infection (*Figure 1H*). Notably, *RBMX2* knockout in EBL cells and silencing in H1299 cells demonstrated significant improvements in cell survival at 96 and 120 hpi compared to wild-type (WT) EBL cells and H1299 ShNC cells, as evidenced by crystal violet staining (*Figure 1—figure supplement 2C*).

## *RBMX2*-regulated genes associated with cell tight junction and EMT-related pathways

To investigate the role of RBMX2 knockout during *M. bovis* infection, we selected RBMX2 knockout EBL cells and WT EBL cells at 0 hpi (2 hpi post-gentamicin, referred to as 0 hpi), 24 hpi, and 48 hpi for RNA-seq analysis. A total of 16,079 genes were identified with a significance threshold of p < 0.05 and a log$_2$(fold change) >2 relative to WT EBL cells. Notably, 42 genes were significantly regulated at all three time points (0, 24, and 48 hpi) (*Figure 2A*). Of these, 11 genes were significantly upregulated, while 31 were significantly downregulated. A heatmap illustrating the expression levels of these genes across all samples is presented, with each row representing a gene and each column representing a sample (*Figure 2A*).

To further explore the functional implications of these differentially expressed genes (DEGs) in RBMX2 knockout EBL cells during infection, Gene Ontology (GO) and Kyoto Encyclopedia of Genes and Genomes (KEGG) pathway analyses were conducted. The most significantly enriched biological processes included epithelial cell differentiation, cell adhesion, and biological adhesion. The most affected cellular components were the extracellular region, extracellular region part, extracellular space, and plasma membrane part. Molecular functions were primarily associated with activin receptor activity, type I (*Figure 2B*). KEGG pathway analysis revealed that these DEGs were involved in several pathways, including ECM–receptor interaction, cGMP–PKG signaling pathway, PI3K–Akt signaling pathway, cytokine–cytokine receptor interaction, and vascular smooth muscle contraction (*Figure 2C*).

Proteomic analysis of 48 hpi protein samples further validated the phenotypes related to cell tight junctions and EMT. GO analysis identified significant enrichment in pathways related to extracellular space, integrin binding, basement membrane, and glucose homeostasis, whereas KEGG analysis highlighted pathways such as cell adhesion molecules, ECM–receptor interaction, and the TGF-beta signaling pathway (*Figure 2D, E*).

We also assessed how varying infection durations influenced RBMX2 knockout EBL cells using GO analysis. At 0 hpi, DEGs were primarily associated with cell junctions, extracellular regions, and cell junction organization (*Figure 2—figure supplement 1A*). At 24 hpi, enriched pathways included basement membrane, cell adhesion, integrin binding, and cell migration (*Figure 2—figure supplement 1B*). By 48 hpi, genes were mainly annotated to epithelial cell differentiation and were negatively regulated during epithelial cell proliferation (*Figure 2—figure supplement 1C*), suggesting that RBMX2 modulates cellular connectivity throughout the stages of *M. bovis* infection.

KEGG analysis revealed distinct enrichment patterns over time. At 0 hpi, genes were enriched in the MAPK signaling pathway, chemical carcinogen–DNA adducts, and chemical carcinogen–receptor activation (*Figure 2—figure supplement 1D*). At 24 hpi, significant enrichment was observed

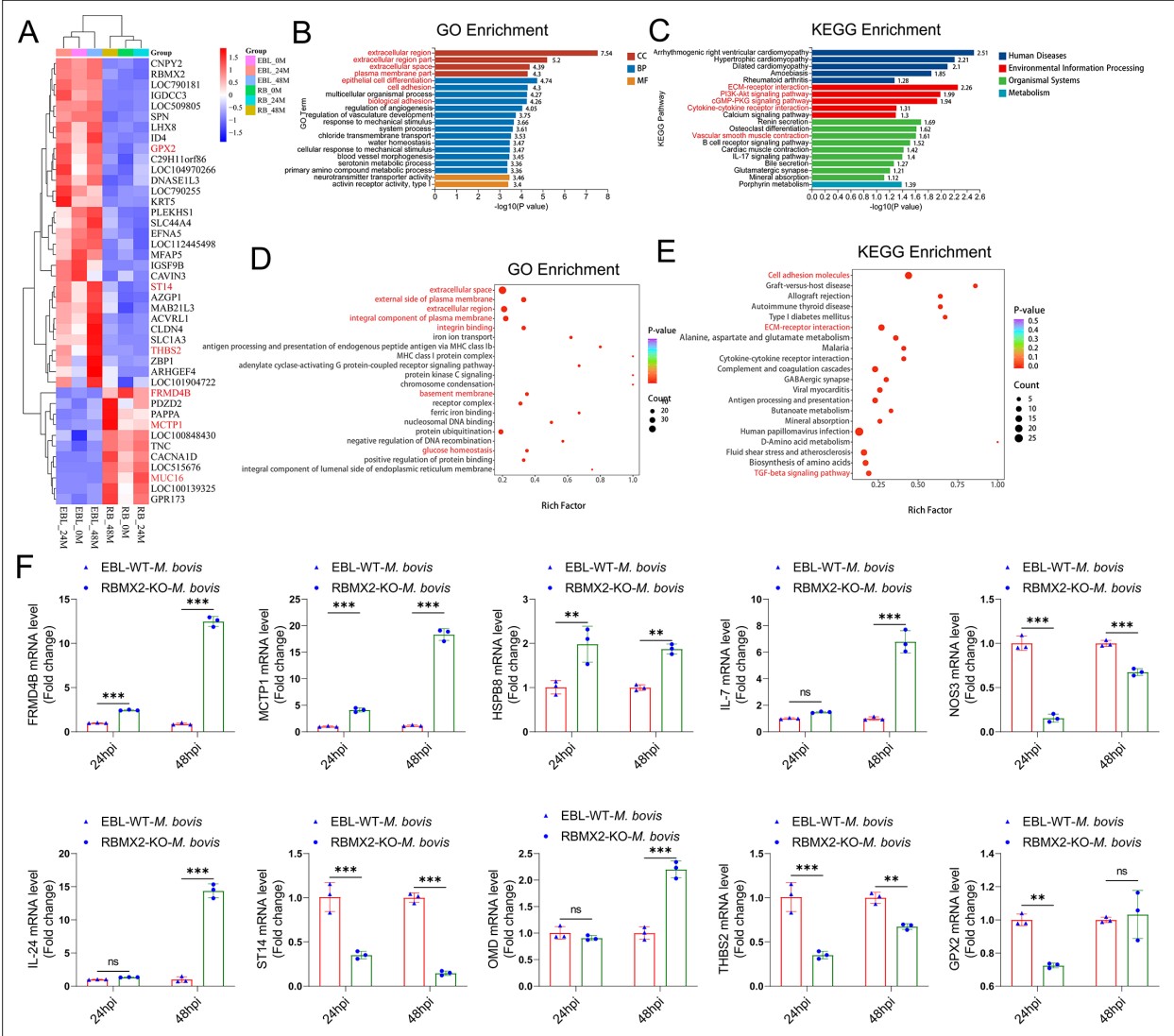

**Figure 2.** Transcriptome and proteomic analysis in *RBMX2* knockout and WT EBL cells after *M. bovis* infection. (**A**) The heatmap illustrates some genes that had been all enriched in RBMX2 knockout and WT EBL cells after *M. bovis* infection in 0 (2 hpi post-gentamicin, recorded as 0 hpi), 24, and 48 hpi. Red represents upregulated genes, and blue represents downregulated genes. Each group represented three independent samples. (**B**) Gene Ontology (GO) analysis of all enriched genes in 0 (2 hpi post-gentamicin, recorded as 0 hpi), 24, and 48 hpi. Data were represented as all enriched pathways in RBMX2 knockout EBL cells relative to WT EBL cells after *M. bovis* infection thrice. (**C**) Kyoto Encyclopedia of Genes and Genomes (KEGG) analysis of all enriched genes in 0 (2 hpi post-gentamicin, recorded as 0 hpi), 24, and 48 hpi. Data were represented as all enriched pathways in RBMX2 knockout EBL cells relative to WT EBL cells after *M. bovis* infection in three times. (**D**) GO analysis of all enriched proteins in 48 hpi. Data were represented as all enriched pathways in RBMX2 knockout EBL cells relative to WT EBL cells after *M. bovis* infection. (**E**) KEGG analysis of all enriched genes in 48 hpi. Data were represented as all enriched pathways in RBMX2 knockout EBL cells relative to WT EBL cells after *M. bovis* infection (MOI 20) in three times. (**F**) Identification of the expression of related genes mRNA enriched by real-time quantitative polymerase chain reaction (RT-qPCR). Data were represented as the fold expression in RBMX2 knockout EBL cells relative to WT EBL cells after *M. bovis* infection. Two-way ANOVA was used to determine the statistical significance of differences between different groups. Ns presents no significance; **$p < 0.01$, and ***$p < 0.001$ indicate statistically significant differences.

The online version of this article includes the following figure supplement(s) for figure 2:

**Figure supplement 1.** Transcriptome analysis in *RBMX2* knockout and WT EBL cells after *M. bovis* infection in different time points.

in ECM–receptor interaction, PI3K–Akt signaling pathway, and focal adhesion (*Figure 2—figure supplement 1E*). At 48 hpi, enrichment was noted in the TGF-beta signaling pathway, transcriptional misregulation in cancer, microRNAs in cancer, small cell lung cancer, and the p53 signaling pathway (*Figure 2—figure supplement 1F*).

To validate the RNA-seq results, we selected 10 genes at random for RT-qPCR analysis (**Figure 2F**, **Figure 3—figure supplement 1C**). The mRNA levels of these genes were consistent with RNA-seq data. These genes included FRMD4B, MCTP1, HSPB8, ST14, and OMD, which are associated with scaffold proteins, cell adhesion, and epithelial barrier function. Additionally, MCTP1, HSPB8, IL-24, IL-7, GPX2, and NOS3 were linked to apoptosis, inflammation, and oxidative stress.

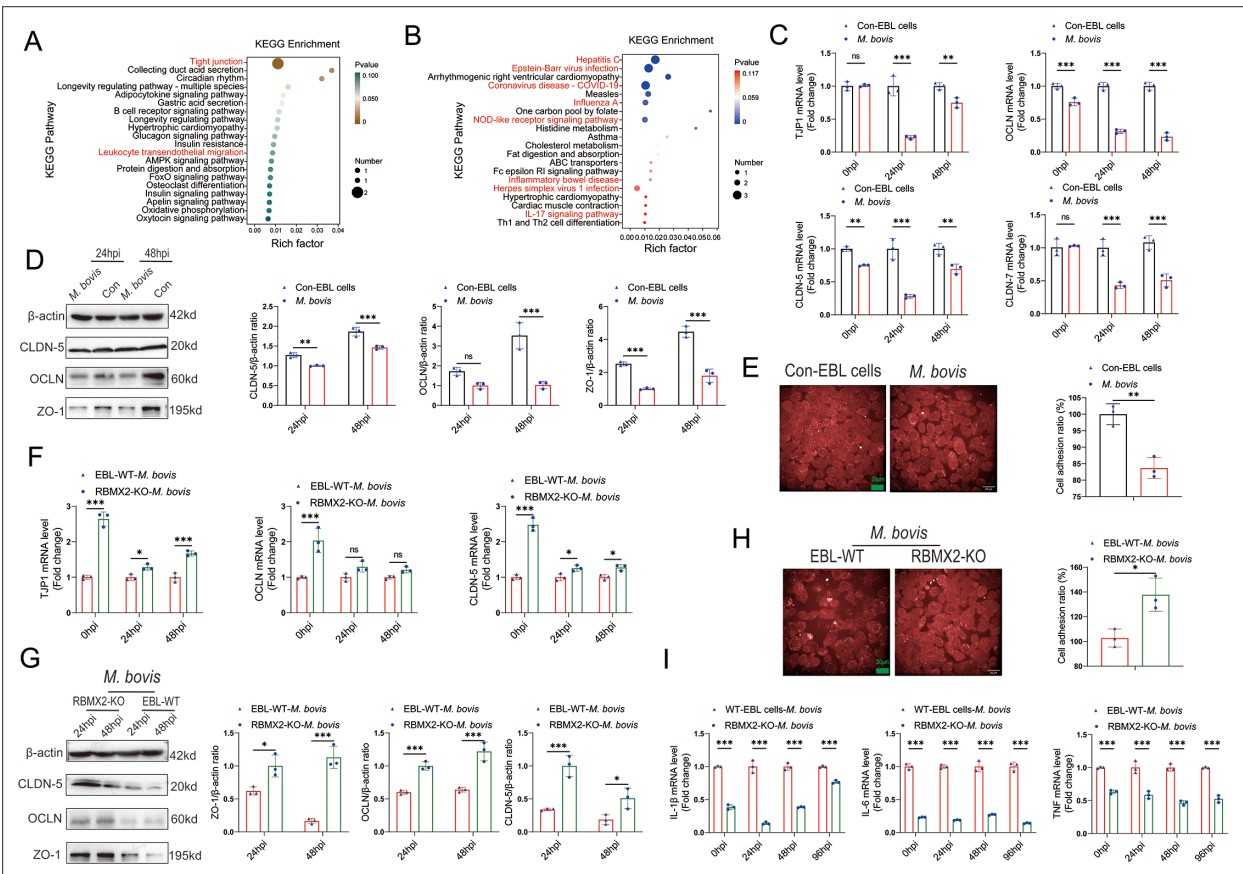

**Figure 3.** *RBMX2* had the potential to induce the disruption of tight junctions in EBL cells after *M. bovis* infection. Kyoto Encyclopedia of Genes and Genomes (KEGG) analysis was conducted to identify the (**A**) downregulated and (**B**) upregulated pathways among the enriched genes after *M. bovis* infection of WT EBL cells. Data were relative to WT EBL cells without *M. bovis* infection. The expression of epithelial cells tight junction-related (**C**) mRNAs (*TJP1*, *CLDN-5*, *CLDN-7*, and *OCLN*) and (**D**) proteins (ZO-1, CLDN-5, and OCLN) were assessed after *M. bovis* infection of WT EBL cells via real-time quantitative polymerase chain reaction (RT-qPCR) and WB. Data were relative to WT EBL cells without *M. bovis* infection. (**E**) Cell adhesion ratio was evaluated via cell adhesion assay after WT EBL cells were infected with *M. bovis* using high-content imaging. Data were relative to WT EBL cells without *M. bovis* infection. Scale bar: 20 µm. (**F, G**) The expression of epithelial tight junction-related (**F**) mRNAs (*TJP1*, *CLDN-5*, and *OCLN*) and (**G**) proteins (ZO-1, CLDN-5, and OCLN) in *RBMX2* knockout EBL cells after *M. bovis* infection through real-time quantitative polymerase chain reaction (RT-qPCR) and WB. Data were relative to WT EBL cells with *M. bovis* infection. (**H**) Cell adhesion assay was conducted to assess the cell adhesion ratio of *RBMX2* knockout EBL cells after infection with *M. bovis*. Data were relative to WT EBL cells with *M. bovis* infection. Scale bar: 20 µm. (**I**) Expression of inflammatory factors-related factors (*IL-6*, *IL-1β*, and *TNF*) was assessed after *RBMX2* knockout EBL cells infected by *M. bovis*. Data were relative to WT EBL cells with *M. bovis* infection. *T*-test and two-way ANOVA were used to determine the statistical significance of differences between different groups. Ns presents no significance; *$p < 0.05$, **$p < 0.01$, and ***$p < 0.001$ indicate statistically significant differences. Data were representative of at least three independent experiments.

The online version of this article includes the following source data and figure supplement(s) for figure 3:

**Source data 1.** Original western blots for panel D, indicating the relevant bands.

**Source data 2.** Original files for western blot analysis displayed in panel D.

**Source data 3.** Original western blots for panel G, indicating the relevant bands.

**Source data 4.** Original files for western blot analysis displayed in panel G.

**Figure supplement 1.** Transcriptomic analysis of *M. bovis*-infected EBL cells.

## *RBMX2* interferes with the integrity of tight junctions in epithelial cells caused by *M. bovis*

Through transcriptome sequencing, we identified 41 genes that underwent transcript changes following *M. bovis* infection. Of these, 22 genes were upregulated, while 19 genes were downregulated (*Figure 3—figure supplement 1A, B*). The downregulated genes were primarily associated with tight junction and leukocyte transendothelial migration (*Figure 3A*). In contrast, the upregulated genes were mainly linked to immunity-related pathways, including the defense response to viruses and regulation of viral processes (*Figure 3B*).

Based on these findings, we hypothesized that *RBMX2* could damage the integrity of epithelial cell tight junctions. To validate this prediction, we constructed a model of *M. bovis*-destroyed tight junctions in EBL cells. EBL cells were infected with *M. bovis*, and we subsequently performed RT-qPCR and western blot (WB) analyses to assess the expression of cell tight junction-related mRNAs (*TJP1*, *OCLN*, *CLDN-5*, and *CLDN-7*) and proteins (ZO-1, OCLN, and CLDN-5). We found that the expression levels of most mRNAs and proteins were significantly reduced following *M. bovis* infection compared to the uninfected control (*Figure 3C, D*).

Tight junctions are critical intercellular adhesion structures that define the permeability of barrier-forming epithelial cells (*Wibbe and Ebnet, 2023*). The ability of epithelial cells to maintain functional intercellular adhesion is essential for forming a tight epithelial protective barrier (*Bhat et al., 2018*). To further investigate the effect of *M. bovis* on epithelial cell adhesion, we conducted a cell adhesion assay, revealing a significant decrease in the intercellular adhesion ratio (*Figure 3E*). These results validated that *M. bovis* can compromise the tight junction of EBL cells.

To explore the impact of *RBMX2* on the epithelial cells barrier, we infected RBMX2 knockout and WT EBL cells with *M. bovis* and assessed the expression of epithelial barrier-related mRNAs (*TJP1*, *OCLN*, and *CLDN-5*) and proteins (ZO-1, OCLN, and CLDN-5) using RT-qPCR and WB. The *RBMX2* knockout EBL cells exhibited a significant upregulation of all three proteins compared to WT EBL cells (*Figure 3F, G*). The cell adhesion assay also demonstrated that *RBMX2* knockout EBL cells displayed a heightened adhesion ratio relative to WT EBL cells (*Figure 3H*).

Tight junctions are essential components of the epithelial barrier and can be compromised by bacterial infections, contributing to inflammation (*Zou et al., 2016*; *Edelblum and Turner, 2009*). To investigate whether *RBMX2* mediates intracellular inflammation by regulating the tightness of the epithelial barrier, we measured the mRNA levels of inflammatory factors (IL-1β, TNF, and IL-6) in *RBMX2* knockout EBL cells post-infection. The results showed a decrease in the expression of these inflammatory factors in RBMX2 knockout EBL cells compared to WT EBL cells after *M. bovis* infection (*Figure 3I*).

In summary, our findings indicate that *RBMX2* affects the integrity of epithelial cell tight junctions, and knocking out *RBMX2* enhances tight junction formation while decreasing the inflammatory response induced by *M. bovis*.

## *RBMX2* disrupted the epithelial barrier via activating *p65*

The MAPK pathway and NF-kappaB pathway play crucial roles in regulating tight junctions between epithelial cells (*Bell and Watson, 2013*; *Kitagawa et al., 2014*; *Lin et al., 2021*; *Jeong et al., 2017*). To further elucidate the regulatory mechanism of *RBMX2* in the epithelial barrier's tight junctions, we found that the MAPK pathway was enriched in our transcriptome sequencing data, and the potential regulatory transcription factor p65 was predicted through the JASPAR database (*Supplementary file 2*).

We assessed the phosphorylation levels of MAPK pathway proteins and p65 in RBMX2 knockout EBL cells, revealing a significant reduction in the phosphorylation of p65 and MAPK/p38/JNK in these cells following *M. bovis* infection compared to WT EBL cells (*Figure 4A, B*).

To determine whether RBMX2 specifically regulates MAPK/p38/JNK proteins and p65 to promote disruption of tight junctions and reduce intercellular adhesion, RBMX2 knockout EBL cells were treated with PMA (a p65 activator), Anisomycin (a JNK activator), and ML141 (a p38 activator) for 12 hr prior to *M. bovis* infection. The optimal concentrations of these activators were determined using Western blotting (WB) and CCK-8 cell viability assays, resulting in concentrations of 100 nM for PMA, 10 μM for Anisomycin, and 10 μM for ML141, with no significant effects on cell viability (*Figure 4—figure supplement 1A–C*). WB results indicated that activation of p65 and the MAPK/p38

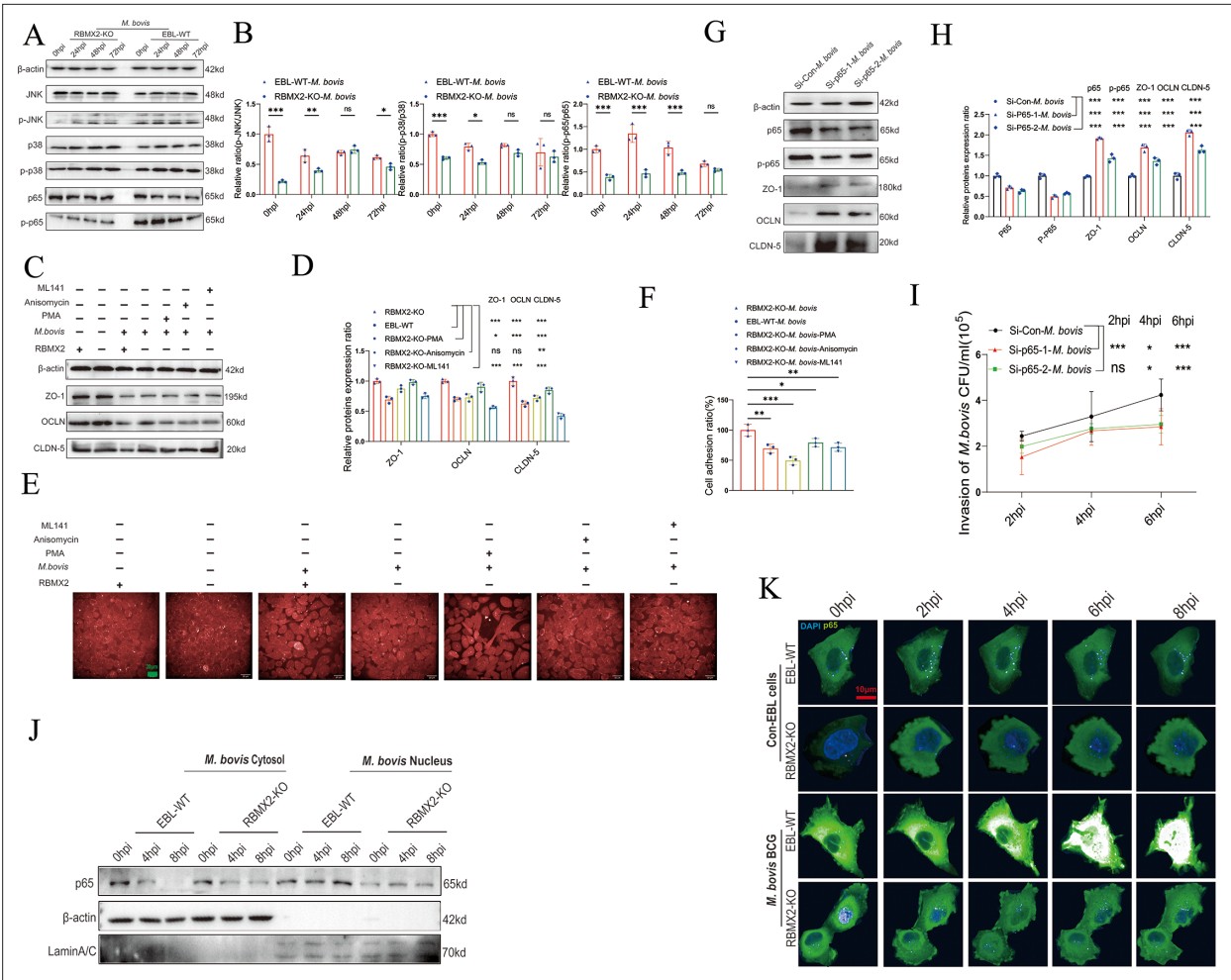

**Figure 4.** *RBMX2* facilitated the disruption of epithelial tight junctions through the promotion of p65 protein phosphorylation and translocation and then enhanced the processes of *M. bovis* adhesion, invasion, and intracellular survival. (**A, B**) Activation of the MAPK pathway-related protein and p65 protein was activated after *RBMX2* knockout and WT EBL cells infected by *M. bovis* via WB. Data were relative to WT EBL cells with *M. bovis* infection. (**C, D**) Expression of tight junction-related proteins (ZO-1, CLDN-5, and OCLN) was assessed in *RBMX2* knockout EBL cells treated with three p38/p65/JNK pathways activators after *M. bovis* infection via WB. Data were relative to *RBMX2* knockout EBL cells untreated activators with *M. bovis* infection. (**E, F**) Evaluate the impact of three *p38/p65/JNK* pathways activators on the ratio of intercellular adhesion via cell adhesion assay. Data were relative to *RBMX2* knockout EBL cells untreated activators with *M. bovis* infection. Scale bar: 20 µm. (**G, H**) Evaluate the silencing efficiency of small interfering RNA (siRNA) on p65 protein expression and its impact on the expression of ZO-1, CLDN-5, and OCLN proteins through WB. Data were relative to siRNA-NC in WT EBL cells with *M. bovis* infection. (**I**) The effect of p65 silencing on the invasive ability of *M. bovis* in WT EBL cells. Data were relative to siRNA-NC in WT EBL cells with *M. bovis* infection. (**H**) The effect of *RBMX2* on the nuclear translocation of p65 protein after *M. bovis* infection using WB. β-Actin presents cytosol and Lamin A/C presents nucleus. Data were relative to RBMX2 knockout EBL cells after *M. bovis* infection. (**K**) The effect of *RBMX2* on the nuclear translocation of p65 protein after *BCG* infection using high-content real-time imaging. Using the pCMV-EGFP-p65 plasmid, transfect RBMX2 knockout and WT EBL cells. The nucleus is stained with blue fluorescence. Data were relative to WT EBL cells without BCG infection. One- and two-way ANOVA were used to determine the statistical significance of differences between different groups. Ns presents no significance; *p < 0.05, **p < 0.01, and ***p < 0.001 indicate statistically significant differences. Data were representative of at least three independent experiments.

The online version of this article includes the following source data and figure supplement(s) for figure 4:

**Source data 1.** Original western blots for panel A, indicating the relevant bands.

**Source data 2.** Original files for western blot analysis displayed in panel A.

**Source data 3.** Original western blots for panel C, indicating the relevant bands.

**Source data 4.** Original files for western blot analysis displayed in panel C.

**Source data 5.** Original western blots for panel G, indicating the relevant bands.

**Source data 6.** Original files for western blot analysis displayed in panel G.

**Source data 7.** Original western blots for panel J, indicating the relevant bands.

*Figure 4 continued on next page*

*Figure 4 continued*

**Source data 8.** Original files for western blot analysis displayed in panel J.

**Figure supplement 1.** The optimal concentration of activators and their impact on cell viability.

**Figure supplement 1—source data 1.** Original western blots for panels A–C, indicating the relevant bands.

**Figure supplement 1—source data 2.** Original files for western blot analysis displayed in panels A–C.

pathways suppressed the expression of tight junction proteins, including ZO-1, OCLN, and CLDN-5, following *M. bovis* infection (*Figure 4C, D*).

To investigate whether RBMX2 regulates the cell adhesion ratio via p65 and MAPK/p38/JNK pathways, we examined the effects of p65, p38, and JNK activators on cell adhesion in RBMX2 knockout EBL cells. Cell adhesion assays demonstrated that treatment with the p65 activator significantly reduced the adhesion ratio compared to treatments with the p38 and JNK activators (*Figure 4E, F*).

To further confirm the association between p65 and tight junction regulation in WT EBL cells after *M. bovis* infection, we used small interfering RNA (siRNA) to knock down p65 expression. WB analysis showed that siRNA effectively reduced both p65 and phosphorylated p65 (p-p65) expression (*Figure 4G, H*). Furthermore, WB results revealed that p65 suppression enhanced the expression of ZO-1, OCLN, and CLDN-5 following *M. bovis* infection (*Figure 4G, H*). Additionally, silencing p65 in WT EBL cells inhibited *M. bovis* invasion into epithelial cells, as demonstrated by plate counting assays (*Figure 4I*).

Finally, to investigate the relationship between *RBMX2*-mediated p65 and the findings above, EBL cells were infected with *M. bovis* and *M. bovis* BCG. WB and high-content live imaging system outcomes demonstrated diminished nuclear translocation of the p65 protein in *RBMX2* knockout EBL cells compared with WT EBL cells (*Figure 4J, K*).

In summary, our findings validate that *RBMX2* can regulate the phosphorylation and nuclear translocation of p65 protein, leading to the degradation of tight junction proteins in EBL cells infected with *M. bovis*.

## *RBMX2* promotes adhesion, invasion, and intracellular survival of pathogens

Disrupting the intercellular tight junction barrier enhances bacterial adhesion and invasion (*Singh and Phukan, 2019*; *Gu et al., 2008*). Based on these findings, we hypothesize that *RBMX2* may facilitate the adhesion and invasion of *M. bovis* by degrading the tight junction proteins in EBL cells. To test this hypothesis, EBL cells were infected with *M. bovis* at an MOI of 20:1, plate count assays indicated a significant reduction in *M. bovis* adhesion in *RBMX2*-KO#17 and KO#23 EBL cells compared with WT EBL cells at 15 min, 30 min, 1 hr, and 2 hr post-infection (*Figure 5—figure supplement 1A*). Furthermore, the invasion of *M. bovis* in *RBMX2*-KO#17 and KO#23 EBL cells was decreased at 2, 4, and 6 hr post-infection compared with WT EBL cells (*Figure 5—figure supplement 1B*). Additionally, the intracellular survival of *M. bovis* in *RBMX2*-KO#17 and KO#23 EBL cells was lower at 24, 48, 72, and 96 hpi compared with WT EBL cells (*Figure 5—figure supplement 1C*). Silencing *RBMX2* in human lung epithelial cells (H1299) also led to a significant reduction in the adhesion, invasion, and intracellular survival of *M. bovis* (*Figure 5—figure supplement 1D–F*). Thus, the host factor *RBMX2* is critical for promoting the adhesion, invasion, and intracellular survival of *M. bovis*. In addition, pathway activators were employed to investigate the relationship between pathway activation and the regulation of *M. bovis* adhesion and invasion by *RBMX2*. The results showed that applying a p65 activator to *RBMX2* knockout EBL cells significantly impaired the tight junction function of the epithelial barrier, thereby enhancing *M. bovis* adhesion and invasion in EBL cells by regulating the phosphorylation and nucleation of p65 (*Figure 5—figure supplement 2A, B*). Hence, RBMX2 promotes *M. bovis* adhesion and invasion in EBL cells by regulating the phosphorylation and nucleation of p65.

To further investigate the role of RBMX2 in adhesion and invasion across different virulent mycobacterial species, we employed both attenuated virulent *M. bovis* BCG and *Mycobacterium smegmatis* to infect EBL cells. Plate count assays demonstrated a significant decrease in the adhesion of both *M. bovis* BCG and *M. smegmatis* in RBMX2 knockout EBL cells at multiple time points post-infection compared to WT EBL cells (*Figure 5—figure supplement 2C, D*). Moreover, the invasion of both mycobacterial species was notably reduced at various hours post-infection in RBMX2 knockout

cells (*Figure 5—figure supplement 2E, F*). These findings suggest that the virulence of *Mycobacterium* does not influence the ability of RBMX2 to facilitate adhesion and invasion in EBL cells.

Additionally, to assess whether RBMX2 regulates interactions with other pathogens associated with bovine pneumonia, EBL cells were infected with Salmonella and *Escherichia coli*. Our results revealed that RBMX2 knockout EBL cells exhibited a reduction in bacterial adhesion and invasion across all tested species compared to WT cells (*Figure 5—figure supplement 2E, F*).

Collectively, these experiments provide strong evidence that RBMX2 broadly enhances the adhesion and invasion of pathogens linked to bovine pneumonia, underscoring its potential as a therapeutic target for developing disease-resistant cattle.

## *RBMX2* is highly expressed in lung cancer and regulates cancer-related metabolites

Previous studies have demonstrated a link between bacterial infection and the initiation of EMT (*Harrandah et al., 2021*; *Li et al., 2021*; *Bessède et al., 2015*). Prolonged infection with *M. tb* or *M. bovis* induces oxidative stress, activates Toll-like receptors, and elicits the release of inflammatory cytokines (*Ferluga et al., 2020*; *Strong et al., 2022*; *Vignesh et al., 2023*). Consequently, this phenomenon creates a favorable microenvironment for tumor development, progression, and dissemination (*Certo et al., 2021*; *Harmon et al., 2020*). Epidemiological investigations have established a correlation between TB and the occurrence of lung cancer (*Rodel et al., 2021*; *Park et al., 2023*; *Hu, 2023*; *Menakuru et al., 2023*). However, the precise cellular mechanism underlying this association remains unclear.

In our research, we conducted a comprehensive analysis of gene families, revealing a remarkable degree of *RBMX2* conservation among bovine, monkey, and human sources (*Figure 5A*). To further elucidate the involvement of *RBMX2* in cancer, we utilized the *TIMER2.0* database to assess its expression patterns across various cancer types within The Cancer Genome Atlas (TCGA). Our findings indicated a notable upregulation of *RBMX2* in lung cancer, specifically in lung adenocarcinoma (LUAD) and lung squamous cell carcinoma (LUSC) (*Figure 5B*).

To investigate the elevated expression of *RBMX2* in lung cancer further, we measured its expression in both lung cancer epithelial cell lines and normal lung epithelial cell lines using RT-qPCR. The results demonstrated that *RBMX2* expression in lung cancer epithelial cell lines (NCIH1299, NCIH460, and CALU1) was significantly higher than that in normal lung epithelial cells (BEAS-2B) (*Figure 5C*). Additionally, immunofluorescence analysis revealed that *RBMX2* expression levels were also higher in LUAD and LUSC tumor tissues compared to adjacent non-cancerous lung tissues (*Figure 5D–F*).

Furthermore, we investigate whether *RBMX2* regulates specific EMT-related metabolites to mediate the EMT process in EBL cells following *M. bovis* infection. Our metabolomic analysis of EBL cell samples at 48 hpi revealed that *RBMX2* knockout primarily enriched pathways related to nucleotide metabolism, biosynthesis of cofactors, biosynthesis of nucleotide sugars, pentose and glucuronate interconversions, vascular smooth muscle contraction, amino sugar, and nucleotide sugar metabolism, chemokine signaling pathway, aldosterone synthesis and secretion, and cGMP–PKG signaling pathway. These pathways are associated with tumor cell proliferation, migration, and invasion (*Figure 5G*). Differential metabolites are related to tumor metabolism, mainly including 6-glucuronic acid estriol, adenosine, uridine 5′-triphosphate, and 5′-deoxy-5′-(methylthio) (*Figure 5H*).

## *RBMX2* promotes the transformation of EBL cells from epithelial cells to mesenchymal cells

To investigate the association between *RBMX2* and EMT induced by *M. bovis*, we initially infected EBL cells directly with *M. bovis*. This infection did not effectively induce the expression of mesenchymal cell marker proteins (*Figure 6—figure supplement 1A*). Previous literature indicates that *M. bovis* infection in macrophages, along with its secreted proteins, can stimulate the production of cytokines such as TNF, IL-6, and TGF-β, all of which promote EMT process (*Liu et al., 2020*; *Shrestha et al., 2020*; *Hao et al., 2019*; *Xu et al., 2007*).

To further explore the impact of *M. bovis* infection on EMT progression in EBL cells, we constructed a coculture model using *M. bovis*-infected BoMac cells (*Figure 6A*, *Figure 6—figure supplement 1B*). In this model, we identified EMT-related cytokines, including IL-6 and TNF, in *M. bovis*-infected BoMac cells, revealing a significant increase compared to uninfected BoMac and EBL cells (*Figure 6B*).

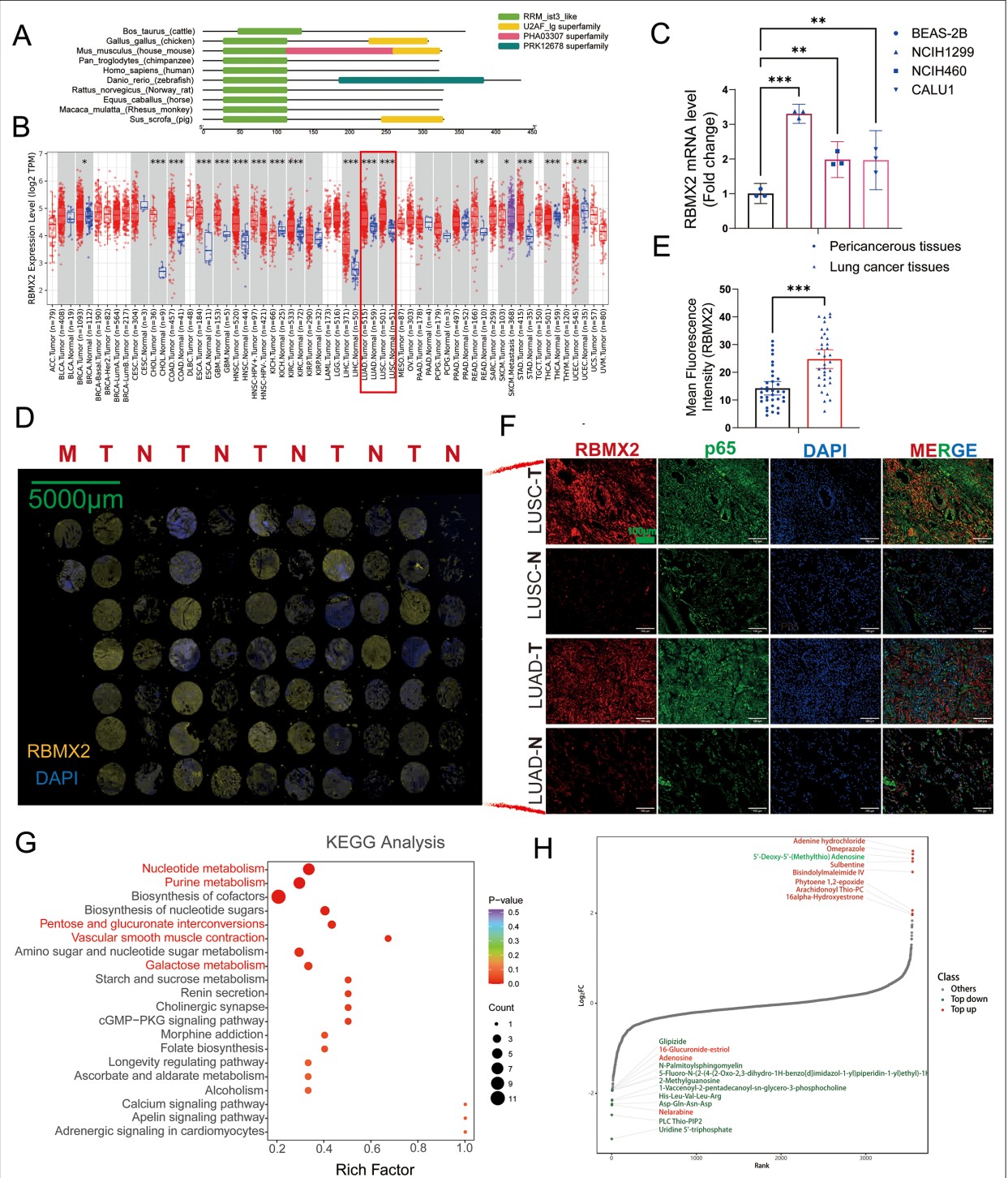

**Figure 5.** *RBMX2* is highly expressed in tumor tissues and regulates cancer-related metabolites. (**A**) A comparative analysis of the functional domains of the *RBMX2* protein across ten different species.(**B**) Analyzing the expression patterns of *RBMX2* in pan-cancer using *TIMER2.0* cancer database. (**C**) The expression of *RBMX2* in different lung cancer cells and normal lung epithelial cells via real-time quantitative polymerase chain reaction (RT-qPCR). Data were relative to normal lung epithelial cells (BEAS-2B). (**D, E**) The expression of *RBMX2* in lung cancer clinical tissues via IF. RBMX2 is stained with yellow fluorescence, and the nucleus is stained with blue fluorescence. Data were relative to pericancerous lung tissues. Scale bar: 5000 µm. (**F**) The expression of *RBMX2* and p65 in lung adenocarcinoma (LUAD) and lung squamous cell carcinoma (LUSC) clinical tissues via IF. *RBMX2* is stained with red fluorescence, p65 is stained with green fluorescence, and the nucleus is stained with blue fluorescence. Data were relative to normal lung tissues. Scale bar: 100 µm.(**G**) Kyoto Encyclopedia of Genes and Genomes (KEGG) analysis of differential metabolite enrichment pathways in *RBMX2* knockout EBL

*Figure 5 continued*

cells compared to WT EBL cells after *M. bovis* infection. (H) Dynamic distribution map of top 20 differential metabolites in RBMX2 knockout EBL cells compared to WT EBL cells after *M. bovis* infection. *p < 0.05, **p < 0.01, and ***p < 0.001 indicate statistically significant differences.

The online version of this article includes the following figure supplement(s) for figure 5:

**Figure supplement 1.** *RBMX2* enhanced the processes of *M. bovis* adhesion, invasion, and intracellular survival.

**Figure supplement 2.** *RBMX2* enhanced the processes of pathogen adhesion, invasion, and intracellular survival.

Notably, *RBMX2* expression was upregulated in EBL cells within this coculture model following *M. bovis* infection (*Figure 6C*).

We then examined the morphology changes in epithelial cells within the coculture model using electron microscopy. The results demonstrated that most cells transitioned from circular to an elongated morphology (*Figure 6D*). Staining of the EBL cells' cytoskeleton with ghost pen cyclic peptide highlighted significant morphological alterations, with EBL cells transforming from spherical to spindle-shaped (*Figure 6—figure supplement 1C*).

Additionally, we assessed the expression of EMT-related mRNAs and proteins in EBL cells after coculture with *M. bovis-infected* BoMac cells. This analysis revealed a significant downregulation of the epithelial cell marker protein *E-cadherin* and a notable upregulation of mesenchymal markers *N-cadherin* and *MMP-9*, as determined by RT-qPCR and WB (*Figure 6E*, *Figure 6—figure supplement 1D*). The coculture with *M. bovis-infected* BoMac cells enhanced the migratory and invasive properties of EBL cells, as demonstrated by Transwell assay (*Figure 6F, G*).

To confirm the impact of *RBMX2* knockout on the EMT process in the coculture model, we observed that *RBMX2* knockout cells exhibited significant upregulation of the epithelial cell marker protein *E-cadherin* and downregulation of the mesenchymal cell marker *N-cadherin* and *MMP-9* compared to WT EBL cells, as shown by RT-qPCR and WB (*Figure 6H*, *Figure 6—figure supplement 1E*). Wound-healing and Transwell assays demonstrated that the migration and invasion rates of *RBMX2* knockout cells were markedly lower than those of WT EBL cells (*Figure 6I–K*).

Finally, we silenced RBMX2 in the human lung cancer epithelial cell line H1299, which expresses high levels of RBMX2, to assess the effect on EMT-related proteins, as well as invasion and migration ability following *M. bovis* infection. The results indicated that RBMX2 significantly inhibited the EMT process in H1299 cells post-infection (*Figure 6—figure supplement 2A, B*).

In conclusion, we successfully established a coculture model involving *M. bovis*-infected BoMac cells that induce EMT in EBL cells, thereby demonstrating that the host factor *RBMX2* effectively promotes the EMT process in this context.

## *RBMX2* facilitates the EMT process via the *p65/MMP-9* pathway

EMT enhances the migration and invasion capabilities of tumor cells (*Na et al., 2020*; *Fedele et al., 2022*; *Alqurashi et al., 2023*). Both p65 and *MAPK/p38/JNK* pathways have been shown to regulate the EMT process through various mechanisms (*Mirzaei et al., 2022*; *O'Leary et al., 2017*; *Stefani et al., 2021*; *Kim et al., 2022*). To elucidate the precise regulatory role of *RBMX2* in the EMT process within the coculture model of EBL cells, we assessed the *MAPK* pathway and p65 protein levels via WB. Our findings revealed a significant reduction in the phosphorylation of *p65* and *MAPK/p38/JNK* in *RBMX2* knockout cells infected with *M. bovis,* compared to WT EBL cells (*Figure 6—figure supplement 2C*).

To identify the specific signaling pathways regulating the EMT phenotype involving p65 and MAPK/p38/JNK in the coculture model, RBMX2 knockout EBL cells were treated with PMA, Anisomycin, and ML141. Our results showed that p65 suppresses E-cadherin expression while enhancing MMP-9 expression (*Figure 7A*). Activation of the MAPK/p38 pathway inhibits E-cadherin expression and promotes N-cadherin expression. In contrast, the MAPK/JNK pathway suppresses N-cadherin expression while enhancing the expression of both E-cadherin and MMP-9. Furthermore, a comprehensive analysis of the relationship between pathway activation and cellular migration and invasion revealed that activation of the p65 pathway increases the migratory and invasive abilities of EBL cells, as demonstrated by Transwell assays (*Figure 7B, C*) and wound-healing assays (*Figure 7D, E*).

Invasion and migration are critical stages in tumor development, influenced by *p65*-dependent factors such as matrix metalloproteinases, urokinase fibrinogen activators, and interleukin-8

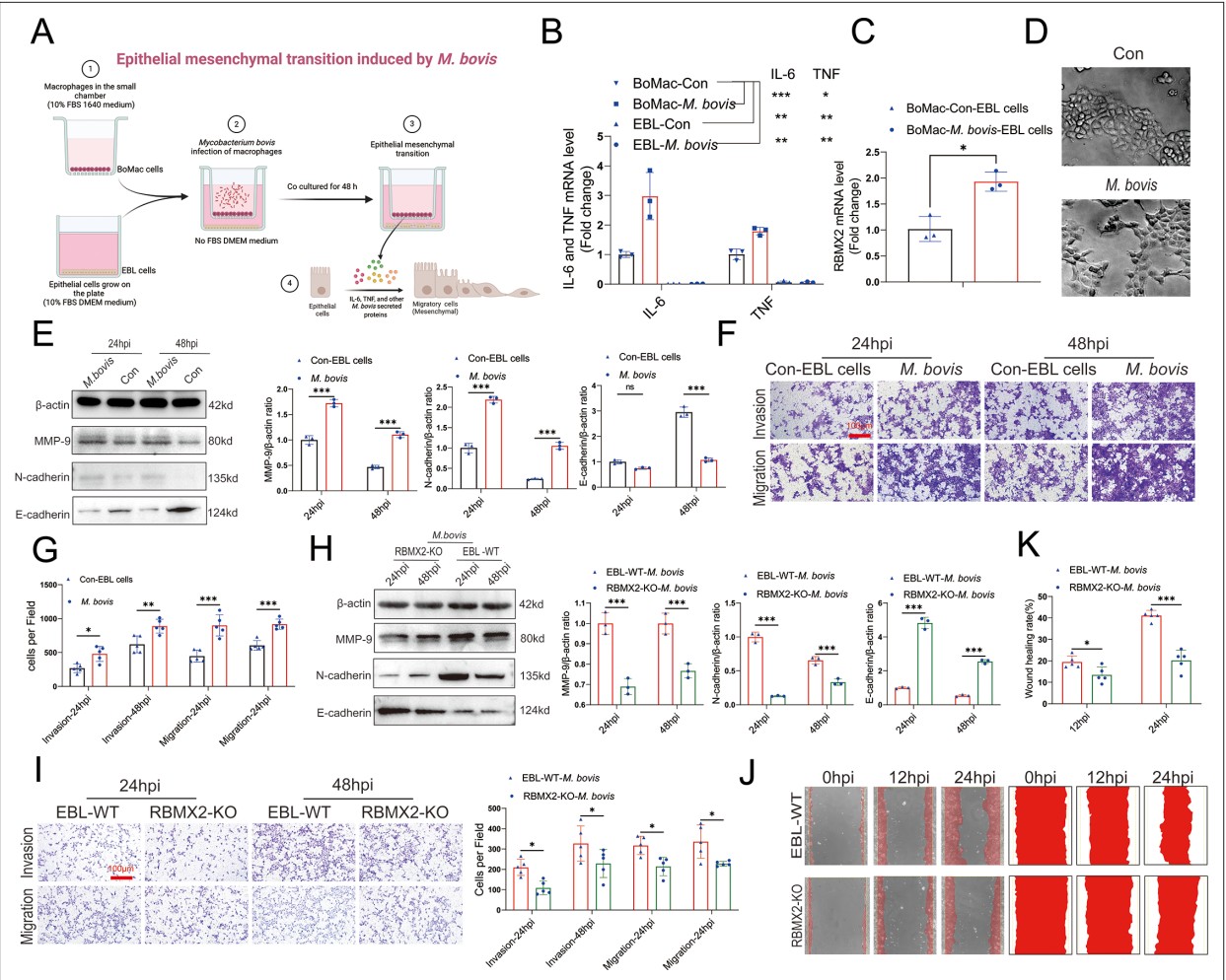

**Figure 6.** *RBMX2* facilitates epithelial–mesenchymal transition (EMT) process in EBL cells after *M. bovis*-infected BoMac cells. (**A**) A pattern diagram illustrated *M. bovis*-infected BoMac cells inducing EMT of EBL cells coculture model, drawing by BioRender. (**B**) Detection of IL-6 and TNF expression levels in EBL cells and BoMac cells infected with *M. bovis* using real-time quantitative polymerase chain reaction (RT-qPCR). Data were relative to BoMac cells without infection of *M. bovis*. (**C**) Detection of RBMX2 expression levels in a coculture model EBL cells after *M. bovis* infection using RT-qPCR. Data were relative to BoMac cells without infection of *M. bovis*. (**D**) Observation of morphological changes in EBL cells infected with *M. bovis* under electron microscopy. (**E**) EMT-related proteins (*MMP-9*, *N-cadherin*, and *E-cadherin*) expression was verified in coculture model EBL cells after *M. bovis* infection through WB. Data were relative to coculture model EBL cells without *M. bovis* infection. (**F, G**) The impact of coculture model EBL cells after *M. bovis* infection on migration and invasion capacity was detected using Transwell assay. Data were relative to coculture model EBL cells without *M. bovis* infection. (**H**) The detection of epithelial–mesenchymal transition (EMT)-related proteins (*MMP-9*, *N-cadherin*, and *E-cadherin*) of *RBMX2* knockout EBL cells after *M. bovis*-infected BoMac cells via WB. Data were relative to WT EBL cells after *M. bovis*-infected BoMac cells. (**I**) The change in the migratory and invasive capabilities of *RBMX2* knockout EBL cells after *M. bovis*-infected BoMac cells was assessed via Transwell assay. Data were relative to WT EBL cells after *M. bovis*-infected BoMac cells. (**J, K**) Validate the changes in migration abilities of *RBMX2* knockout EBL cells after *M. bovis*-infected BoMac cells through wound-healing assay. Data were relative to WT EBL cells after *M. bovis*-infected BoMac cells. *T*-test and two-way ANOVA were used to determine the statistical significance of differences between different groups. Ns presents no significance; *p < 0.05, **p < 0.01, and ***p < 0.001 indicate statistically significant differences. Data were representative of at least three independent experiments.

The online version of this article includes the following source data and figure supplement(s) for figure 6:

**Source data 1.** Original western blots for panel E, indicating the relevant bands.

**Source data 2.** Original files for western blot analysis displayed in panel E.

**Source data 3.** Original western blots for panel H, indicating the relevant bands.

**Source data 4.** Original files for western blot analysis displayed in panel H.

**Figure supplement 1.** *RBMX2* facilitates epithelial–mesenchymal transition (EMT) process in EBL cells after *M. bovis*-infected BoMac cells.

**Figure supplement 1—source data 1.** Original western blots for panel A, indicating the relevant bands.

*Figure 6 continued on next page*

*Figure 6 continued*

**Figure supplement 1—source data 2.** Original files for western blot analysis displayed in panel A.

**Figure supplement 1—source data 3.** Original western blots for panel B, indicating the relevant bands.

**Figure supplement 1—source data 4.** Original files for western blot analysis displayed in panel B.

**Figure supplement 2.** *RBMX2* facilitates epithelial–mesenchymal transition (EMT) in H1299 cells.

**Figure supplement 2—source data 1.** Original western blots for panel A, indicating the relevant bands.

**Figure supplement 2—source data 2.** Original files for western blot analysis displayed in panel A.

**Figure supplement 2—source data 3.** Original western blots for panel C, indicating the relevant bands.

**Figure supplement 2—source data 4.** Original files for western blot analysis displayed in panel C.

(*Gonzalez-Avila et al., 2020*; *Cox, 2021*; *Kwaan and Lindholm, 2019*; *Mego et al., 2015*; *Kim, 2020*). To further substantiate the correlation between the p65 protein and the regulation of the EMT process in EBL cells post-*M. bovis* infection, we employed siRNA to inhibit p65 expression in WT EBL cells. This suppression was found to inhibit *MMP-9* expression, as demonstrated by WB (*Figure 7F, G*).

To thoroughly verify the regulatory mechanism between RBMX2 and p65, we initiated our investigation by conducting an in-depth analysis of the RBMX2 promoter region to identify potential interactions with the transcription factor p65 (*Supplementary file 2*). Initially, we performed molecular docking simulations to predict the binding affinity and interaction patterns between RBMX2 and p65 proteins. These simulations revealed multiple amino acid residues within the RBMX2 protein that formed strong, stable interactions with p65. The docking analysis yielded a high docking score of 1978.643 (*Figure 7K*), indicating a significant likelihood of a direct physical interaction between these two proteins.

To complement the protein-protein interaction analysis, we next investigated whether p65 could directly bind to the promoter region of the RBMX2 gene at the transcriptional level. Using the JASPAR database, a comprehensive resource for transcription factor-binding profiles, we queried the RBMX2 promoter sequence for potential p65-binding sites. This analysis identified several putative binding motifs, suggesting that p65 may act as a transcriptional regulator of RBMX2 expression.

To experimentally validate this transcriptional regulatory relationship, we employed a dual-luciferase reporter assay. We cloned the RBMX2 promoter region containing the predicted p65-binding sites into a luciferase reporter plasmid. This construct was then co-transfected into cultured cells along with a plasmid expressing p65. The luciferase activity was significantly increased in cells expressing p65 compared to control groups, providing functional evidence that p65 enhances the transcriptional activity of the RBMX2 promoter (*Figure 7I*).

Furthermore, to confirm the direct binding of p65 to the RBMX2 promoter in a chromatin context, we performed chromatin immunoprecipitation followed by quantitative PCR (ChIP-qPCR). In this assay, we used specific antibodies against p65 to immunoprecipitate chromatin fragments containing p65-bound DNA. The enriched DNA fragments were then analyzed using primers targeting the RBMX2 promoter region. Our results demonstrated a significant enrichment of the RBMX2 promoter in the p65 immunoprecipitated samples compared to the IgG control, thereby confirming that p65 physically associates with the RBMX2 promoter in vivo (*Figure 7J*). Collectively, these findings—ranging from computational docking predictions to transcriptional reporter assays and ChIP validation—provide strong evidence supporting a direct regulatory interaction between p65 and RBMX2. This regulatory mechanism may play a critical role in the biological pathways involving these two molecules, particularly in contexts such as inflammation, immune response, or cellular stress, where p65 (a subunit of NF-κB) is known to be prominently involved.

Moreover, MMP-9 is known to induce EMT in epithelial cells, enhancing their invasiveness. The regulatory mechanisms involving p65 and *MMP-9* have been documented in several studies (*Zuo et al., 2011*; *Peng et al., 2023*). In our research, we identified multiple binding sites in the promoter sequences of p65 and MMP-9 through molecular docking (*Figure 7L*) and confirmed the binding site between *MMP-9* promoter and *p65* protein via ChIP-PCR (*Figure 7M*). Using protein docking, we validated the relationship between bovine p65 protein and MMP-9, with a binding score of 1784.378 (*Figure 7N*).

In summary, *RBMX2* facilitates the EMT process of EBL cells following *M. bovis* infection by activating the *p65/MMP-9* pathway.

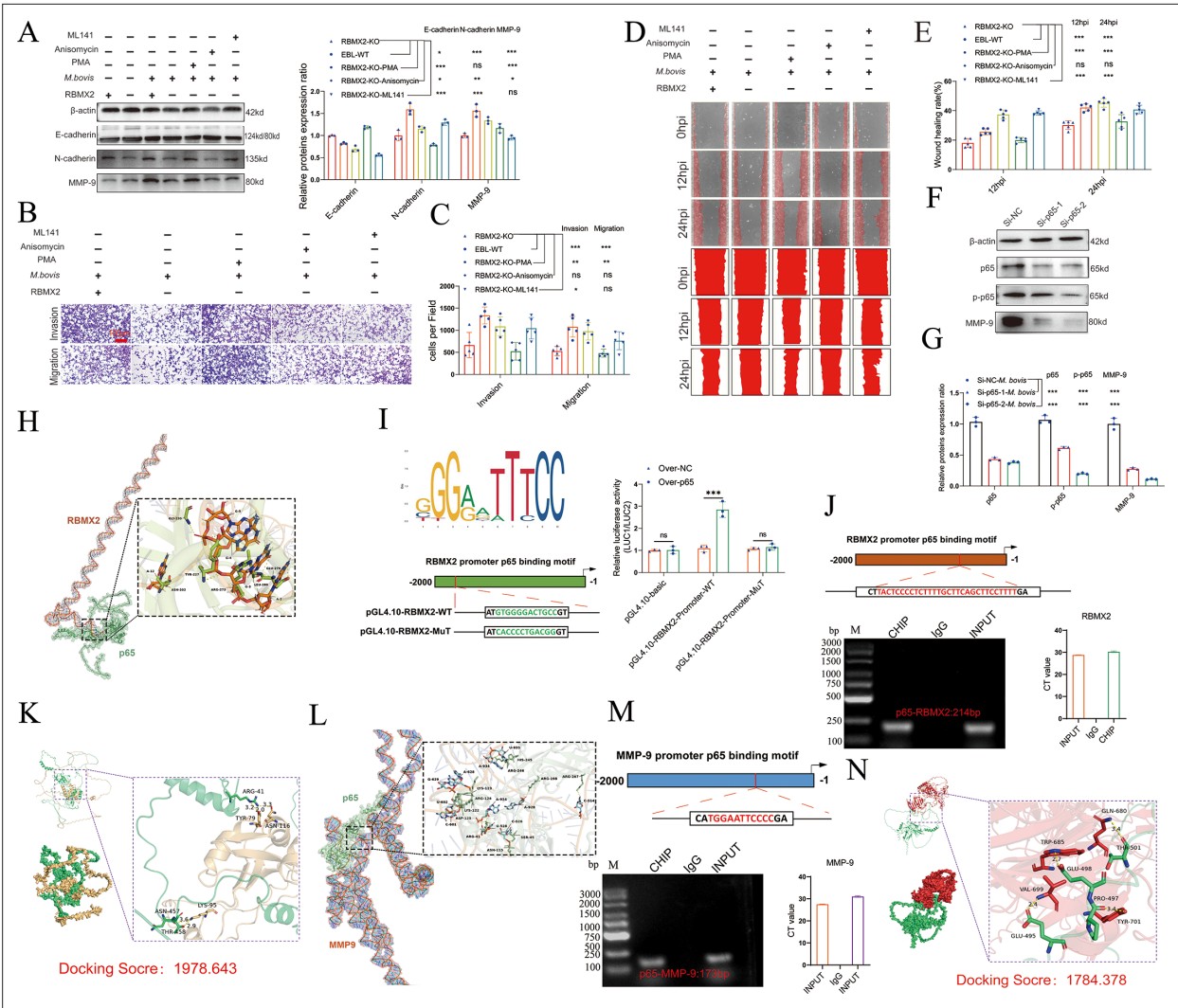

**Figure 7.** *RBMX2* facilitates the epithelial–mesenchymal transition (EMT) in EBL cells via *p65/MMP-9* pathway. (**A**) Evaluate the impact of pathway activations on expression of EMT-associated proteins (*MMP-9, N-cadherin*, and *E-cadherin*) in *RBMX2* knockout EBL cells after *M. bovis-infected* BoMac cells. Data were relative to *RBMX2* knockout EBL cells untreated activators. (**B, C**) Evaluate the impact of pathway activations on the migratory and invasive capabilities of RBMX2 knockout EBL cells after *M. bovis-infected* BoMac cells via Transwell assay. Data were relative to *RBMX2* knockout EBL cells untreated activators. (**D, E**) Evaluate the impact of pathway activations on the migratory capabilities of *RBMX2* knockout EBL cells after *M. bovis-infected* BoMac cells via wound-healing assay. Data were relative to *RBMX2* knockout EBL cells untreated activators. (**F, G**) The impact of p65 silencing on the expression of *MMP-9* protein in WT EBL cells after *M. bovis* infection was assessed using WB. Data were relative to siRNA-NC in WT EBL cells with *M. bovis* infection. (**H**) Predicting the binding ability of RBMX2 promoter and p65 using molecular docking dynamics. (**I**) Verification of *RBMX2* promoter region and p65 interaction using dual luciferase reporter system. (**I**) Using p65 antibody to precipitate p65 protein in EBL cells, and verification of *RBMX2* promoter region and p65 interaction using ChIP-PCR assay. (**K**) Predicting potential binding sites for p65 and *RBMX2* via protein docking. (**L**) Verification of MMP-9 promoter region and p65 interaction using dual luciferase reporter system. (**M**) Verification of MMP-9 promoter region and p65 interaction using ChIP-PCR. (**N**) Predicting potential binding sites for p65 and *MMP-9* via protein docking. Two-way ANOVA was used to determine the statistical significance of differences between different groups. *p < 0.05, **p < 0.01, and ***p < 0.001 indicate statistically significant differences. Data were representative of at least three independent experiments.

The online version of this article includes the following source data for figure 7:

**Source data 1.** Original western blots for panel A, indicating the relevant bands.

**Source data 2.** Original files for western blot analysis displayed in panel A.

**Source data 3.** Original western blots for panel F, indicating the relevant bands.

**Source data 4.** Original files for western blot analysis displayed in panel F.

## Discussion

The resurgence of *M. bovis*-associated TB presents a substantial global challenge, affecting both livestock and human populations. Notably, *M. bovis* exhibits over 99% nucleotide similarity with *M. tb* (*Reis and Cunha, 2021*). The pathogenesis of TB is a complex process influenced by a confluence of bacterial, host, and environmental factors. Both *M. tb* and *M. bovis* have evolved sophisticated strategies to evade host immune responses, enabling their long-term intracellular persistence. It is estimated that approximately one-third of the global population harbors latent TB infections (*Fan et al., 2018*; *Borkowska et al., 2017*).

The successful establishment of infection by mycobacteria is contingent upon their ability to circumvent early innate immune defenses, employing both transcriptional and post-transcriptional regulatory strategies within host macrophages (*Rastogi et al., 2023*; *Corleis and Dorhoi, 2020*; *Abebe, 2021*; *Reuschl et al., 2017*; *Dorhoi and Kaufmann, 2014*). Despite notable advancements in the field, unraveling the mechanisms of immune evasion and understanding the dynamics of latent infections caused by *M. tb* and *M. bovis* remains a substantial challenge. The cellular immune response to these pathogens is inherently complex, further compounded by the microdiversity of mycobacteria within individual hosts and the variability of immune responses among different individuals (*Abebe, 2021*; *Rajaram et al., 2014*).

Lung epithelial cells serve as the primary physical barrier against infection and play a pivotal role in the innate immune response. These cells not only function as a frontline defense but also recruit and activate antigen-presenting cells, such as macrophages, to initiate adaptive immune responses against *M. bovis* infection (*D'Agnillo et al., 2021*; *Barros et al., 2022*). Given that alveolar epithelial cells can act as reservoirs for *M. bovis*, their role in the infection process is particularly significant. Additionally, activated macrophages and neutrophils enhance the bactericidal effects of alveolar epithelial cells, further contributing to the host's defense mechanisms (*Hu et al., 2023*; *Buckley and Turner, 2018*).

In our study, we employed a CRISPR–Cas9 mutant library generated in our laboratory to explore potential host factors involved in *M. bovis* infection. Notably, we identified RBMX2 as a protein that significantly promotes *M. bovis* infection. The *RBMX2* protein, characterized by an RNA recognition motif within its 56–134 amino acid residue region, is implicated in mRNA splicing through spliceosomes and is emerging as a potential molecular marker for sperm activity (*Ahmadi Rastegar et al., 2015*). The downregulation of RBMX2 in the X chromosome of lung telocytes suggests its involvement in cellular immunity (*Zhu et al., 2015*).

Further exploration of RBM genes in cattle revealed that EIF3G, RBM14, RBM42, RBMX44, RBM17, PUF60, SART3, and RBM25 belong to the same subfamily as RBMX2. Previous studies have demonstrated that genes within this subfamily can regulate the proliferation and lifecycle of cancer cells. For instance, EIF3G modulates the mTOR signaling pathway, inhibiting the proliferation and metastasis of bladder cancer cells (*Zhang et al., 2021*). Similarly, RBM14 has been linked to the reprogramming of glycolysis in lung cancer, acting as a novel epigenetically activated oncogene (*Hu et al., 2023*). Despite these insights, the specific function of RBMX2 in the context of *M. bovis* infection and its potential role in cancer pathogenesis remains largely unexplored.

Our findings reveal that *RBMX2* is upregulated in TB-infected cells, demonstrating its capacity to enhance *M. bovis* infection. Transcriptomic analyses suggest that *RBMX2* may disrupt tight junctions within epithelial cells and promote EMT following *M. bovis* infection.

Epithelial cells serve as more than passive barriers; they actively participate in innate immunity by regulating cytokine secretion and maintaining barrier integrity (*Buckley and Turner, 2018*). Disruptions in epithelial tight junctions, often exacerbated by pro-inflammatory stimuli from pathogenic bacteria, can facilitate bacterial translocation and subsequent infection (*Buckley and Turner, 2018*; *Savagner, 2015*; *Kyuno et al., 2021*).

In our experiments, we observed that the disruption of the epithelial barrier facilitated *M. bovis* adhesion and invasion. Conversely, the knockout of RBMX2 stabilized the epithelial barrier, attenuating *M. bovis* invasion and intracellular survival. This stabilization also mitigated downstream innate immune responses, reducing cellular inflammation, ROS production, and apoptosis.

The loss of tight junction integrity is a precursor to EMT, a process increasingly recognized for its role in cancer progression (*Li et al., 2021*; *Savagner, 2015*; *Kyuno et al., 2021*; *Hashimoto and Oshima, 2022*). Recent studies have established a link between bacterial infections and EMT induction, particularly in the context of gastric adenocarcinomas (*Malfertheiner et al., 2023*; *Brito*

*et al., 2019*). Epidemiological evidence suggests that TB may serve as a risk factor for lung cancer (*Ho and Leung, 2018*; *Christopoulos et al., 2014*); however, the underlying cellular mechanisms remain elusive. Chronic TB infection has been implicated in lung carcinogenesis, with reports indicating that BCG vaccination can enhance the survival of tumor cells under inflammatory conditions (*Nalbandian et al., 2009*; *Holla et al., 2014*). There is growing epidemiological evidence suggesting that chronic TB infection represents a potential risk factor for the development of lung cancer. Studies have shown that individuals with a history of TB exhibit a significantly increased risk of lung cancer, particularly in areas of the lung with pre-existing fibrotic scars, indicating that chronic inflammation, tissue repair, and immune microenvironment remodeling may collectively contribute to malignant transformation (*Hwang et al., 2022*). Moreover, EMT not only endows epithelial cells with mesenchymal features that enhance migratory and invasive capacity but is also associated with the acquisition of cancer stem cell-like properties and therapeutic resistance (*Brabletz, 2012*). Therefore, EMT may serve as a crucial molecular link connecting chronic TB infection with the malignant transformation of lung epithelial cells, warranting further investigation in the intersection of infection and tumorigenesis.

Moreover, *M. tb*-infected THP-1 cells have been shown to induce EMT in LUAD epithelial cells (*Gupta et al., 2016*). Chronic infection with *M. tb* is associated with oxidative stress and inflammatory cytokine production, fostering an environment conducive to tumor progression (*Leung et al., 2020*; *de la Barrera et al., 2004*). Our analysis of RBMX2 across various cancers revealed increased expression levels in LUAD and LUSC, suggesting a conserved role in tumor biology across species.

In light of these findings, we constructed a model of *M. bovis* infection in EBL cells to investigate EMT induction. Initial results indicated that *M. bovis* alone did not induce EMT; however, a coculture model incorporating *M. bovis*-infected BoMac cells successfully induced EMT in EBL cells. Notably, the knockout of RBMX2 in this context inhibited EMT, suggesting that RBMX2 may elevate the risk of lung cancer through EMT induction following *M. bovis* infection. Meanwhile, metabolic pathways enriched after RBMX2 knockout, such as nucleotide metabolism, nucleotide sugar synthesis, and pentose interconversion, primarily support cell proliferation and migration during EMT by providing energy precursors, regulating glycosylation modifications, and maintaining redox balance; cofactor synthesis and amino sugar metabolism participate in EMT regulation through influencing metabolic remodeling and extracellular matrix interactions; chemokine and cGMP–PKG signaling pathways may further mediate inflammatory responses and cytoskeletal rearrangements, collectively promoting the EMT process.

In summary, RBMX2 drives TB pathogenesis by compromising epithelial barriers and inducing EMT. Targeting RBMX2 may present a promising avenue for the prevention and treatment of TB in both humans and animals. Additionally, the effective modulation of *RBMX2* could potentially mitigate the incidence of TB-associated EMT and its implications for lung cancer development in the near future.

## Conclusion

Our research findings indicate that RBMX2 significantly enhances the invasive capacity of *M. bovis* by promoting the activation and nuclear translocation of the p65 protein. This activation compromises the integrity of the epithelial cell barrier and induces EMT through the p65/MMP-9 signaling pathway. These results highlight the crucial role of RBMX2 in *M. bovis* pathogenesis and underscore its potential as a therapeutic target for preventing infection-related complications.

## Methods

### Patients and lung tissues

This study was approved by the Ethics Committee of Inner Mongolia Autonomous Region People's Hospital, and written informed consent was obtained from all participating patients (Approval Number: 2020021). A total of 35 specimens of lung cancer tissue, along with adjacent normal lung tissue, were collected. We randomly selected the above specimens, including age and gender. All patients involved in this study had not received any medication, chemotherapy, or radiation therapy prior to surgical resection. The samples were subsequently prepared for immunohistochemistry and fluorescence in situ hybridization assays.

## Cell lines

EBL and 293T cells were generously provided by M. Heller from Friedrich-Loeffler-Institute. These cells were cultured in heat-inactivated 10% fetal bovine serum (FBS) supplemented with Dulbecco's modified Eagle medium (DMEM, Gibco, USA) at 37°C and 5% $CO_2$ (*Chen et al., 2018*). BoMac cells were kindly provided by Judith R. Stabel from the Johne's Disease Research Project at the United States Department of Agriculture in Ames, Iowa, and were maintained according to previously established protocols (*Zhao et al., 2021*). BoMac cells were grown in heat-inactivated 10% FBS supplemented with Roswell Park Memorial Institute 1640 (RPMI 1640, Gibco, USA) at 37°C and 5% $CO_2$. Bovine lung alveolar primary cells were isolated in our laboratory and cultured in heat-inactivated 10% FBS supplemented with DMEM (Gibco, USA) at 37°C and 5% $CO_2$. BEAS-2B, NCIH1299, NCIH460, and CALU1 epithelial cells were all provided by Professor Lei Shi from Lanzhou University, and all culture procedures were conducted according to standardized protocols. We conduct regular testing on all cell lines to eliminate contamination from mycoplasma and black fungus.

## Bacterium

*M. bovis* (ATCC:19210), originally isolated from a cow, is maintained in this laboratory. *M. smegmatis* mc (*Preda et al., 2023*) 155 (NC_008596.1) and *M. bovis* BCG-Pasteur (ATCC:35734) were generously provided by Professor Luiz Bermudez from Oregon State University. All strains were cultured in a Middlebrook 7H9 broth (BD, MD, USA) supplemented with 0.5% glycerol (Sigma, MO, USA), 10% oleic acid–albumin–dextrose–catalase (OADC, BD, USA) and 0.05% Tween 80 (Sigma, MO, USA) or on Middlebrook 7H11 agar plates (BD, MD, USA) containing 0.5% glycerol (Sigma, MO, USA) and 10% OADC (BD).

Prior to infection, the optical densities of bacterial cultures at 600 nm (OD600) were adjusted to the required MOI using the standard turbidimetric card. The cultures were then centrifuged at $3000 \times g$ for 10 min. The precipitated bacteria were resuspended in medium and dispersed using an insulin syringe. Subsequently, 50 µl of 10-fold serially diluted bacterial suspension was plated onto Middlebrook 7H11 agar to determine the number of viable bacteria (colony-forming units, CFUs). All experiments involving *M. bovis* were conducted in strict accordance with the biosafety protocols established for the Animal Biosafety Level 3 Laboratory of the National Key Laboratory of Agricultural Microbiology at Huazhong Agricultural University (*Peng et al., 2024a*; *Peng et al., 2024b*).

*E. coli* (ATCC:25922) was donated by Professors Zhou Rui and Wang Xiangru of Huazhong Agricultural University (*Xu et al., 2023*), while *Salmonella* (ATCC:14028) has been preserved and passed down through generations in our laboratory. These strains were resuscitated, and single colonies were purified, cultured, and grown in Luria–Bertani broth.

## Generation of the *RBMX2*-KO EBL cells

The small guide RNA (sgRNA) sequence targeting the bovine RBMX2 gene (5′-GAATGAGCGTGA GGTCGAAC-3′) was cloned into the lentiviral vector pKLV2-U6gRNA5(BbsI)-PGKpuro2ABFP (#67991), kindly provided by Professor Zhao Shuhong from Huazhong Agricultural University, to generate the recombinant lentivirus. EBL cells were infected with either the RBMX2-targeting lentivirus or the empty vector lentivirus as a negative control. At 48–60 hpi, puromycin (2.0 mg/ml) was added to select for stably transduced cells. Positive clones were subsequently enriched, and monoclonal cell lines were obtained through limiting dilution. Successful knockout of RBMX2 was confirmed by PCR analysis.

| sgRNA | sequence |
|---|---|
| sgRNA-F | 5′-CCTTGCCCAATTTTTCGGAGG-3′ |
| sgRNA-R | 5′-ACAGGAGGATGGTAGTAACGG-3′ |

## Extraction of total RNA and RT-qPCR

Cold phosphate-buffered saline (PBS, HyClone, China) was used to wash the cells three times, after which 1 ml of Trizol (Invitrogen, USA) was added per well to lyse the cells. The lysate was collected in EP tubes, and 200 µl of chloroform was subsequently added. The mixture was vortexed for 30 s and then centrifuged at 12,000 rpm for 10 min at 4°C. Following centrifugation, 500 µl of the supernatant was transferred to a new EP tube. To this supernatant, 500 µl of isopropanol was added and mixed

gently by inversion. The mixture was allowed to stand for 10 min at 4°C before being centrifuged again at 12,000 rpm for 15 min at 4°C. The supernatant was discarded, revealing the RNA pellet. Next, the RNA pellet was washed with 1 ml of 75% ethanol and centrifuged at 7500 rpm for 5 min at 4°C. The supernatant was removed, and the RNA pellet was air-dried for 15 min. Subsequently, 20 μl of DEPC-treated water was added, and the mixture was incubated at 58°C in a water bath for 10 min to dissolve the RNA. Purified RNA was then obtained.

To assess RNA purity, the OD260/OD280 ratio was measured using a NanoDrop ND-1000 instrument (Agilent Inc, USA), with values between 1.8 and 2.0 indicating acceptable purity. RNA integrity and potential contamination with genomic DNA were evaluated using denaturing agarose gel electrophoresis. The samples were stored at –80°C for later analysis.

Reverse transcription of the RNA samples was performed using HiScript III RT SuperMix for qPCR (+gDNA wiper, Vazyme, China). Four microliters of 4× gDNA wiper mix were added to 1 μg of RNA, followed by the addition of 16 μl of RNase-free ddH$_2$O. The mixture was incubated at 42°C for 2 min to eliminate genomic DNA. Reverse transcription was subsequently carried out by adding 4 μl of 5× HiScript III qRT SuperMix, with the mixture incubated at 37°C for 15 min and then at 85°C for 5 s to synthesize cDNA.

cDNA expression across different sample groups was quantified using AceQ qPCR SYBR Green Master Mix (Vazyme, China) in a ViiA7 real-time PCR machine (Applied Biosystems Inc, USA). The final volume of each real-time PCR reaction was 20 μl, comprising 10 μl of 2× AceQ qPCR SYBR Green Master Mix, 0.4 μl of upstream primer (10 μM), 0.4 μl of downstream primer (10 μM), 0.4 μl of 50× ROX reference dye 2, 3 μl of cDNA template, and 5.8 μl of ddH$_2$O. The PCR conditions were as follows: 95°C for 5 min (1 cycle), 95°C for 10 s (40 cycles), and 60°C for 30 s (40 cycles). The primer sequences for RT-qPCR are provided in *Supplementary file 3*.

## Western blot

RIPA reagent (Sigma, USA), supplemented with protease inhibitors and phosphatase inhibitors (Roche, China), was added to the cell samples, and the cells were lysed on ice for 30 min. The supernatant was collected by centrifugation at 12,000 rpm and 4°C for 10 min. Protein concentrations of each sample were determined using a BCA kit (Beyotime, China), and the proteins were adjusted to equal concentrations.

The protein samples were mixed with 5× protein loading buffer and boiled for 10 min. Proteins were then separated using SDS–PAGE on 10% polyacrylamide gels at 100 V for 90 min, followed by transfer to polyvinylidene difluoride (PVDF) membranes (Millipore, Germany) at 100 V for 70 min. The PVDF membrane was incubated in 5% skimmed milk for 4 hr to block non-specific binding. The membrane was washed three times with Tris-buffered saline containing 0.15% Tween-20 (TBST) for 5 min each.

Subsequently, the membrane was incubated with a primary antibody at 4°C for 12 hr, followed by a secondary antibody at room temperature for 1 hr. The following primary antibodies were utilized: ZO-1 (Bioss, China), OCLN (Proteintech, China), CLADN-5 (Proteintech, China), MMP-9 (Proteintech, China), E-cadherin (Proteintech, China), N-cadherin (Bioss, China), LaminA/C (Proteintech, China), β-actin (Bioss, China), p65 (CST, USA), p-p65 (CST, USA), and an MAPK-related antibody (CST, USA). The following secondary antibodies were used: anti-mouse HRP secondary antibody and anti-rabbit HRP (Invitrogen, USA).

## Phylogenetic tree construction and gene structure and function predictive analysis of the bovine RBM family

The bovine RBM protein sequences were converted into FASTA format files and imported into MEGA-X for comparative analysis. The neighbor-joining method was employed to construct the phylogenetic tree of the CsGRF protein. The iTOL website (https://itol.embl.de/) was utilized to enhance the visualization of the CsGRF phylogenetic tree. Additionally, the MEME Suite database (http://meme-suite.org/) was used for the online analysis of conserved motifs within the RBM gene family.

## Adhesion assay

EBL cells were infected with different live bacteria at an MOI of 20 for durations of 15 min, 30 min, 1 hr, and 2 hr at 4°C. Following infection, the cells were washed three times with PBS (Gibco, USA)

to remove extracellular bacteria. After washing, cells were lysed using 0.1% Triton X-100 (Beyotime, China). The resulting lysates were plated onto 7H11 or LB agar plates and incubated for several days to enable enumeration of CFUs.

## Invasion assay

EBL cells were infected with various live bacteria at an MOI of 20 for durations of 2, 4, and 6 hr at 37 °C. After infection, gentamicin (Procell, China) was added at a concentration of 100 µg/ml for 2 hr to eliminate extracellular bacteria. The infected cells were then washed three times with PBS (Gibco, USA) to remove residual extracellular bacteria. Following thorough washing, the cells were lysed using 0.1% Triton X-100 (Beyotime, China). The resulting lysates were plated onto 7H11 or LB agar plates and incubated for several days to allow for CFU enumeration.

## Intracellular survival assay

EBL cells were infected with various live bacterial strains at an MOI of 20 for 6 hr at 37°C. After infection, extracellular bacteria were removed by treating the cells with gentamicin (100 µg/ml; Procell, China) for 2 hr. The infected cells were then washed three times with PBS (Gibco, USA) to eliminate any remaining extracellular bacteria, marking the 0 hr time point.

Following this, the infected cells were cultured in complete medium supplemented with 0.1% Triton X-100 (Beyotime, China) and harvested at selected time points: 0, 24, 48, and 72 hpi. Cell lysates were plated onto 7H11 or LB agar plates and incubated for several days to allow for CFU enumeration.

## RNA-seq, proteomics, and metabolomics

We collected cells at different time points after infection, where the cells at 0 hr were treated with gentamicin 2 hr post-infection with *M. bovis* to eliminate extracellular bacteria. Total RNA was isolated using Trizol Reagent (Invitrogen Life Technologies), and the concentration, quality, and integrity were determined using a NanoDrop spectrophotometer (Thermo Scientific). Three micrograms of RNA were used as input material for RNA sample preparation. Sequencing libraries were generated according to the following steps: mRNA was purified from total RNA using poly-T oligo-attached magnetic beads. Fragmentation was performed using divalent cations under elevated temperature in an Illumina proprietary fragmentation buffer. First-strand cDNA was synthesized using random oligonucleotides and Super Script II. Second-strand cDNA synthesis was subsequently performed using DNA Polymerase I and RNase H. The remaining overhangs were converted into blunt ends via exonuclease/polymerase activities, and the enzymes were removed. After adenylation of the 3′ ends of the DNA fragments, Illumina PE adapter oligonucleotides were ligated for hybridization. To select cDNA fragments of the preferred length of 400–500 bp, the library fragments were purified using the AMPure XP system (Beckman Coulter, Beverly, CA, USA). DNA fragments with ligated adaptor molecules at both ends were selectively enriched using the Illumina PCR Primer Cocktail in a 15-cycle PCR reaction. The products were purified (AMPure XP system) and quantified using the Agilent high-sensitivity DNA assay on a Bioanalyzer 2100 system (Agilent). The sequencing library was then sequenced on the NovaSeq 6000 platform (Illumina) by Shanghai Personal Biotechnology Co, Ltd.

RNA sequencing services were provided by Personal Biotechnology Co, Ltd, Shanghai, China. The data were analyzed using the free online platform Personalbio GenesCloud (https://www.genescloud.cn). Proteomics and metabolomics were performed by METWARE.

## Differential expression analysis of the transcriptome

The statistical methods of HTSeq (0.9.1) RRID:SCR_005514 were used to compare read count values for each gene, which served as raw measurements of gene expression, while FPKM (Fragments Per Kilobase of transcript per Million mapped reads) was used for data normalization. Subsequently, differential gene expression analysis was performed using DESeq (1.30.0) RRID:SCR_000154 under the following screening criteria: $|log_2FoldChange| > 2$ indicated a significant fold change in expression, and a p-value <0.05 was considered statistically significant. The resulting list of DEGs was used for further functional enrichment and pathway analysis.

For clustering analysis of DEGs, the Pheatmap (1.0.8) package in R was used to perform bidirectional clustering analysis on the expression patterns of all DEGs across various samples. The heatmap was constructed based on the expression levels of the same gene across different samples and the

expression patterns of different genes within the same sample. Distance calculations were based on Euclidean distance, and complete linkage clustering was used to visualize overall expression differences among samples and similarities in gene expression patterns.

Moreover, to further enhance the flexibility and reproducibility of data analysis, a localized analysis workflow was introduced. All raw sequencing data (FASTQ files) underwent quality control (using FastQC) and were aligned (using STAR or HISAT2) to the reference genome, followed by gene expression quantification and differential analysis using locally installed HTSeq and DESeq2 pipelines. This local workflow not only improves the transparency and controllability of data processing but also enhances data privacy protection, making it particularly suitable for handling sensitive or large-scale datasets. All analysis parameters were thoroughly documented to ensure experimental reproducibility and result traceability.

Finally, data analysis was also supported by the free online analytical platform Personalbio GenesCloud (https://www.genescloud.cn), which provides a graphical interface and efficient cloud computing resources, supporting differential analysis, functional annotation (GO/KEGG), gene set enrichment analysis, and visualization. The results from local and cloud-based analyses complement each other, ensuring comprehensive and accurate data interpretation.

## GO and KEGG enrichment analysis

We mapped all genes to terms in the GO database and calculated the number of differentially enriched genes for each term. GO enrichment analysis was conducted on the DEGs using the topGO package, where p-values were determined using the hypergeometric distribution method. A threshold of p-value <0.05 was set to identify significant enrichment, allowing us to pinpoint GO terms associated with significantly enriched differential genes and to elucidate the primary biological functions they perform. Additionally, we employed ClusterProfiler (3.4.4) software to conduct KEGG pathway enrichment analysis for the differential genes, concentrating on pathways with a p-value <0.05. The data analysis was carried out using the free online platform Personalbio GenesCloud (https://www.genescloud.cn).

## Flow cytometry assay

Cells were washed with cold PBS and then centrifuged at 1500 rpm for 5 min; the supernatant was discarded. The cells were fixed in 70% ethanol for 12 hr. Following fixation, the cells were centrifuged again at 1500 rpm for 5 min, and the supernatant was removed. The cells were washed once with cold PBS.

For staining, a Cell Cycle and Apoptosis Analysis Kit (Beyotime, China) was utilized. The staining mixture was prepared with a total volume of 535 µl, comprising 500 µl of staining buffer, 25 µl of PI dye, and 10 µl of RNase A. The samples were incubated with the staining mixture at 37°C for 30 min prior to detection.

## Cell viability assay

EBL cells were infected with *M. bovis* at an MOI of 100. Cell viability was assessed at 48, 72, 96, 120, and 144 hpi using the CCK-8 assay (Dojindo, Kumamoto, Japan).

At each time point, fresh medium containing 10% CCK-8 was added to a 96-well cell plate and incubated for 60 min at 37°C. The cell viability was then determined by measuring the absorbance at 450 nm. The percentage of cell viability was calculated using the following formula:

cell viability (%) = [A (infection group) − A (blank group)] / [A (infection group) A (blank group)] × 100

## Wound-healing assay

A wound-healing assay was performed to evaluate the migratory capacity of treated cells following experimental intervention. Briefly, after a 48-hr coculture of EBL cells with BoMac cells post-*M. bovis* infection, the cells were harvested and dissociated using 1x trypsin. Approximately $2 \times 10^5$ cells were then seeded into each well of a 6-well culture plate and allowed to adhere until they reached confluence and formed a uniform monolayer.

Once a confluent cell monolayer was achieved, a standardized 'wound' was created by gently scratching the surface of the monolayer along a straight edge using a 200-µl pipette tip. To remove non-adherent and detached cells, the monolayer was rinsed three times with PBS (Gibco, USA). Subsequently, serum-free culture medium was added to each well to minimize cell proliferation and to better assess cell migration.

Phase-contrast images of the wound area were captured at defined time points—0, 12, and 24 hr—using an inverted microscope. The wound closure rate was quantitatively analyzed using ImageJ software by measuring the change in wound width over time. The percentage of wound healing was calculated relative to the initial wound width at time 0.

This assay provided insights into the effects of *M. bovis* infection and subsequent treatments on cell migration, a critical component of the wound-healing process.

## Cell adhesion assay

To evaluate the impact of RBMX2 knockout on the adhesive properties of EBL cells following *M. bovis* infection, a cell adhesion assay was performed using a commercially available cell adhesion detection kit (Bestbio, China), following the manufacturer's instructions.

Matrigel (20 mg/l), used to simulate an extracellular matrix environment, was coated onto the wells of a 96-well plate at a volume of 100 µl per well. Subsequently, $3 \times 10^4$ cells (from either RBMX2 knockout or WT EBL cell lines) were seeded into each well and incubated for 6 hr in a $CO_2$ incubator at 37°C to allow cell attachment.

Following incubation, non-adherent cells were gently removed by washing the wells twice with PBS. Adherent cells were then fixed and stained with 10 µl of cell stain solution A per well and incubated for an additional 2 hr at 37°C.

The absorbance was measured at 560 nm using a microplate reader to quantify the intensity of staining, which correlates with the number of adhered cells. The results were used to compare the adhesive capacities of RBMX2 knockout versus WT EBL cells under the experimental conditions.

This assay provided a quantitative assessment of cell-matrix adhesion, offering insights into the potential role of RBMX2 in modulating cell adhesion during *M. bovis* infection.

## Transwell assay

To evaluate the migratory and invasive potential of treated EBL cells, a Transwell assay was performed using Transwell chambers (Corning, USA). A total of $5 \times 10^4$ cells suspended in 200 µl of serum-free culture medium were seeded into the upper chamber, while 550 µl of complete medium containing FBS was added to the lower chamber to serve as a chemoattractant.

The cells were allowed to migrate through the membrane for 24 hr in a $CO_2$ incubator at 37°C.

Following incubation, the Transwell insert was carefully removed, and the cells on the upper surface of the membrane were gently wiped away with a cotton swab. The migrated cells on the lower surface were then fixed by incubation with 550 µl of 4% paraformaldehyde (Beyotime, China) for 30 min at room temperature. After fixation, the cells were stained with 550 µl of crystal violet solution (Beyotime, China) for an additional 30 min to allow for visualization of the migrated cells.

The stained membrane was rinsed briefly with PBS for 30 s to remove excess dye, air-dried upside down, and then carefully removed from the insert. It was subsequently mounted onto a glass slide and sealed with neutral balsam to preserve the staining.

Images were captured under a light microscope, and three random fields of view were selected for analysis. The number of migrated cells was quantified using ImageJ software (RRID:SCR_003070). The results were used to compare the migratory abilities of different experimental groups.

This assay provided a quantitative assessment of cell migration, which is essential for understanding the functional impact of treatments or genetic modifications on cell motility.

## Crystal violet staining assay

EBL cells were infected with *M. bovis* at an MOI of 100. The surviving cells on six-well plates were then fixed with 4% paraformaldehyde (Beyotime, China). After fixation, the cells were stained with 0.1% crystal violet staining solution (Beyotime, China) and rinsed five times with PBS (Gibco, USA). The plates were placed in a 37°C oven for 6 hr before being photographed.

### *M. bovis* infection of BoMac and coculture with EBL cells

BoMac cells were seeded into the upper chamber of a 0.4 µm pore size Transwell (Corning, USA) at a density of $3 \times 10^4$ cells per insert in RPMI 1640 medium supplemented with 10% FBS. The following day, the cells were infected with *M. bovis* at an MOI of 5 for 4 hr. After infection, the cells were washed three times with 1× warm PBS and treated with Gentamycin (100 µg/ml) for 2 hr to eliminate any extracellular bacilli.

The infected and uninfected BoMac cells were then incubated for 24 hr to mitigate the effects of *M. bovis* infection. Afterward, the cells underwent an additional 24-hr incubation period in fresh RPMI 1640 medium before coculturing with EBL cells.

In parallel, EBL cells were seeded at a density of $2 \times 10^5$ cells per well onto 12-well plates containing DMEM supplemented with 10% FBS. The following day, culture inserts containing either *M. bovis*-infected or uninfected BoMac cells were introduced into the wells of the 12-well plates containing EBL cells, and the coculture was incubated for up to 24 and 48 hr in serum-free DMEM.

### siRNA transfection

EBL cells were transfected with p65 siRNA (Tsingke, China) using jetPRIME (Polyplus, France) following the manufacturer's instructions. Scrambled siRNA (Tsingke, China) served as a negative control.

Cells were seeded in 12-well plates and cultured at 37°C in a 5% $CO_2$ atmosphere. To prepare the transfection mixture, 0.8 µg of jetPRIME was incubated with 1 µl of siRNA for 10 min. This mixture was then added to the cells and incubated for 12 hr.

Following the incubation, RNA extraction was performed, and the efficiency of transfection was calculated.

The siRNA sequences used were as follows:

> Cattle p65-1 siRNA, sense 5'-GCAGUUUGAUACCGAUGAA (dT)(dT)-3';
> Cattle p65-1 siRNA, antisense 5'-UUCAUCGGUAUCAAACUGC (dT)(dT)-3';
> Cattle p65-2 siRNA, sense 5'-GGACGUACGAGACCUUCAA (dT)(dT)-3';
> Cattle p65-2 siRNA, antisense 5'-UUGAAGGUCUCGUACGUCC (dT)(dT)-3'.

### P65 nuclear translocation assay

For the p65 nuclear translocation assay, RBMX2 knockout and WT cell lines were transfected with the pCMV-EGFP-p65 plasmid. Following transfection, bovine lung epithelial cells were infected with *M. bovis*.

After infection, the cell nuclei were stained with Hoechst dye. The entry of p65 into the nucleus was analyzed at different time points using a confocal high-resolution cell imaging analysis system from PerkinElmer Life and Analytical Sciences Ltd (Britain). This imaging technique allowed for the assessment of p65 translocation into the nucleus in response to infection.

### ChIP-PCR

Formaldehyde cross-linking and ultrasonic fragmentation of cells were performed. One plate of cells was removed, and the volume of the culture medium was measured. 37% formaldehyde was added to achieve a final concentration of 1%, and the plates were incubated at 37°C for 10 min. Glycine (2.5 M) was then added to the plates at a final concentration of 125 mM to terminate cross-linking. After mixing, the culture medium was left at room temperature for 5 min, and the cells were cleaned three times with cold PBS. The cells were scraped off with PBS, centrifuged at 2000 × g for 5 min, and the supernatant was removed. IP lysis solution containing a protease inhibitor was added to lyse the cells (the amount of lysis solution depending on the amount of cell precipitation), and the mixture was fully lysed on ice for 30 min, with the cells being repeatedly blown with a gun or shaken on a vortex mixer to ensure complete lysis.

After sonication, the mixture was centrifuged at 12,000 rpm at 4°C for 10 min to remove insoluble substances and collect the supernatant. 90 µl of the input was retained, and the rest was stored at −80°C. To confirm the presence of the target protein in the sample, 40 µl of the ultrasonic crushing product was taken, mixed with 10 µl of L5 reduced protein loading buffer, and heated for denaturation before performing Western blot detection.

For the remaining 50 µl of the product, 5 µl of Protease K and 2 µl of 5 M NaCl (final NaCl concentration of 0.2 M) were added and incubated at 55°C overnight for cross-linking. After cross-linking, the nucleic acid concentration was measured, and a portion of the sample was taken for PCR amplification, followed by agarose gel electrophoresis to detect the effectiveness of ultrasonic fragmentation and confirm the presence of the target DNA. After verifying the input result, 100 µl of the ultrasonic crushing products was frozen at −80°C.

Next, 900 µl of ChIP Diffusion Buffer containing 1 mM PMSF and 20 µl of 50% of 1× PIC was added, along with an additional 60 µl of Protein A+G Agarose/Salmon Sperm DNA. The mixture was stirred at 4°C for 1 hr, allowed to stand at 4°C for 10 min to precipitate, and then centrifuged at 4000 rpm for 5 min. The sample was divided into two 1.5 ml EP tubes; the target protein IP antibody was added to one tube, while IgG (1 µg of the corresponding species) was added to the other. The samples were shaken overnight at 4°C for precipitation and cleaning of immune complexes.

After the overnight incubation, 200 µl of Protein A+G Agarose/Salmon Sperm DNA was added to each tube, shaken at 4°C for 2 hr, allowed to stand at 4°C for 10 min, and then centrifuged at 4000 rpm for 1 min. The supernatant was removed, and 8 µl of 5 M NaCl and 20 µl of Protein K were added for overnight cross-linking at 55°C. Using databases like JASPAR to predict transcription factor-binding sites, primers were designed and synthesized based on these binding sites. RT-PCR was employed to verify the binding, and after amplification, the products were taken for gel electrophoresis to confirm the correct fragment size (*Supplementary file 4*).

## Dual-luciferase reporter assay

293T cells in the logarithmic growth phase were adjusted to a density of $5 \times 10^4$ cells/mL and inoculated into 48-well cell culture plates, with 300 cells per well. Each concentration gradient was represented by three replicate wells. On the second day after plating, when the cells reached a density of approximately 60–70%, transfection was performed with the following experimental groups: pGL4.10 RRID:Addgene_72684 basic+pRL TK+Over NC, pGL4.10 basic+pRL TK+Over p65, pGL4.10-*RBMX2*-WT+pRL TK+Over NC, pGL4.10-*RBMX2*-WT+pRL TK+Over p65, pGL4.10-*RBMX2*-MUT+pRL TK +Over NC, pGL4.10-*RBMX2*-MUT+pRL TK +Over p65. Transfection complexes were prepared, and dual-luciferase detection was carried out 48 hr post-transfection.

For cell lysis, the cell culture medium was removed, and the cells were gently rinsed twice with PBS. Then, 50 µl of 1× PLB lysis solution was added to each well, and the plates were placed on a shaking table at room temperature for 15 min to ensure complete lysis.

From each sample, 20 µl was taken for measurement. To assess Firefly luciferase activity, 100 µl of Firefly luciferase detection reagent (LAR II reagent, Promega, USA) was added, mixed well, and the relative light unit (RLU1) was measured. After this, 100 µl of Sea Kidney luciferase detection reagent (1× Stop&Glo Reagent, Promega, USA) was added, mixed thoroughly, and the relative light unit (RLU2) was measured.

The activation level of the target reporter genes was compared between different samples by calculating the ratio of RLU1 from the Firefly luciferase assay to RLU2 from the Sea Kidney luciferase assay.

## Protein docking

Molecular docking simulations were conducted to predict the formation of stable complexes between proteins RBMX2 or MMP9 with p65. The aligned sequences of RBMX2, MMP9, and p65 proteins were sourced from the UniProt RRID:SCR_002380 database. Their three-dimensional structures were predicted using AlphaFold and refined by constructing the structures in Avogadro. The structures were then optimized using the MMFF94 force field and exported in PDB format for further optimization in Gauss09.

The docking of the proteins was performed using Hdock, where each protein was treated separately as the receptor and ligand to assess their interactions (*Li et al., 2025*). The resulting docking affinities were annotated to provide insight into the binding interactions. Finally, PyMOL RRID:SCR_000305 was utilized to visualize the binding interaction geometries, allowing for a detailed examination of the molecular interactions between the proteins.

## Statistical analysis

Statistical analysis was performed on all assays conducted in triplicate, with data expressed as the mean ± standard error of the mean. Each experiment was independently repeated three times. GraphPad Prism 7.0 RRID:SCR_002798 (La Jolla) was utilized for statistical analysis, employing a two-tailed unpaired $t$-test with Welch's correction for comparisons between two groups. For comparisons among multiple groups, one- or two-way ANOVA was applied, followed by the LSD test. Statistical significance was indicated at four levels: not significant (ns presents $p > 0.05$), $*p < 0.05$, $**p < 0.01$, and $***p < 0.001$.

## Acknowledgements

We would like to thank the National Key Laboratory of Agricultural Microbiology Core Facility for assistance in high-throughput microscopy, and we are grateful to Zhe Hu for his support of the sample preparation. This work was supported by the Major projects of agricultural biological breeding in China (2023ZD0405003), the National Natural Science Foundation of China (32072942), China Agriculture Research System of MOF and MARA (CARS-37), National Natural Science Foundation of China (81960772), the 'Zhiyuan Talent' Project of Inner Mongolia Medical University (ZY20241102), and Key Project of Inner Mongolia Medical University (YKD2022ZD016).

## Additional information

### Funding

| Funder | Grant reference number | Author |
| --- | --- | --- |
| Major projects of agricultural biological breeding | 2023ZD0405003 | Yingyu Chen |
| National Natural Science Foundation of China | 32072942 | Aizhen Guo |
| China Agriculture Research System of MOF and MARA | CARS-37 | Aizhen Guo |
| National Natural Science Foundation of China | 81960772 | Hongxin Yang |
| Inner Mongolia Medical University | 'Zhiyuan Talent' Project ZY20241102 | Hongxin Yang |
| Inner Mongolia Medical University | Key Project YKD2022ZD016 | Hongxin Yang |

The funders had no role in study design, data collection, and interpretation, or the decision to submit the work for publication.

### Author contributions

Chao Wang, Conceptualization, Data curation, Formal analysis, Supervision, Funding acquisition, Validation, Investigation, Visualization, Methodology, Writing – original draft, Project administration, Writing – review and editing; Yongchong Peng, Data curation, Validation, Investigation, Methodology, Writing – review and editing; Hongxin Yang, Resources, Supervision, Funding acquisition, Validation, Investigation, Visualization, Methodology, Writing – original draft, Writing – review and editing; Yanzhu Jiang, Visualization, Writing – original draft; Abdul Karim Khalid, Data curation, Investigation; Kailun Zhang, Formal analysis, Methodology; Shengsong Xie, Resources, Formal analysis, Visualization, Writing – original draft; Luiz Bermudez, Formal analysis, Supervision, Investigation, Methodology, Writing – original draft, Writing – review and editing; Yong Yang, Resources, Validation, Methodology; Lei Zhang, Investigation, Methodology, Writing – original draft; Huanchun Chen, Validation, Methodology; Aizhen Guo, Resources, Data curation, Formal analysis, Supervision, Funding acquisition, Validation, Visualization, Methodology, Writing – original draft, Project administration, Writing – review and editing; Yingyu Chen, Resources, Data curation, Formal analysis, Supervision, Funding acquisition,

Validation, Investigation, Visualization, Methodology, Writing – original draft, Project administration, Writing – review and editing

### Author ORCIDs
Lei Zhang ![ORCID] https://orcid.org/0000-0002-8566-6068
Yingyu Chen ![ORCID] https://orcid.org/0000-0002-1200-5314

### Ethics

This study was approved by the Ethics Committee of Inner Mongolia Autonomous Region People's Hospital, and written informed consent was obtained from all participating patients (Approval Number: 2020021). A total of 35 specimens of lung cancer tissue, along with adjacent normal lung tissue, were collected. We randomly selected the above specimens, including age and gender. All patients involved in this study had not received any medication, chemotherapy, or radiation therapy prior to surgical resection. The samples were subsequently prepared for immunohistochemistry (IHC) and fluorescence in situ hybridization (FISH) assays.

Reviewer #1 (Public review): https://doi.org/10.7554/eLife.107132.3.sa1
Reviewer #3 (Public review): https://doi.org/10.7554/eLife.107132.3.sa2
Author response https://doi.org/10.7554/eLife.107132.3.sa3

## Additional files

### Supplementary files

Supplementary file 1. *RBMX2* exhibits high homology in amino acid sequence alignment across different species.

Supplementary file 2. P65 with promoter.

Supplementary file 3. Real-time quantitative polymerase chain reaction (RT-qPCR) primers.

Supplementary file 4. ChIP-PCR primers.

MDAR checklist

### Data availability

The sequencing datasets presented in this study can be found on BioProject.

The following dataset was generated:

| Author(s) | Year | Dataset title | Dataset URL | Database and Identifier |
|---|---|---|---|---|
| Chao W, Chen Y | 2024 | Transcriptome sequencing of 48 bovine-derived cell samples | https://www.ncbi.nlm.nih.gov/bioproject/PRJNA1105205/ | NCBI BioProject, PRJNA1105205 |

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
