## [Editor Report · eLife Assessment]

The identification of RBMX2 as a novel regulator linking mycobacterial infection to Epithelial-Mesenchymal Transition and cancer progression are **fundamental** findings that advance our understanding of a major research question about the link between infectious and non-infectious diseases, microbiology and oncology. It does so by introducing RBMX2 as a novel host factor, a potential therapeutic target and biomarker for both TB and lung cancer. The evidence provided is **convincing** because it is appropriate and the validated multi-omics methodologies used are in line with the current state of the art. This study will be of interest to scientists working in the fields of drug discovery, microbiology and oncology.

---

## [Referee Report · Reviewer #3 (Public review)]

Summary:

This study investigates the role of the host protein RBMX2 in regulating the response to *Mycobacterium bovis* infection and its connection to epithelial-mesenchymal transition (EMT), a key pathway in cancer progression. Using bovine and human cell models, the authors have wisely shown that RBMX2 expression is upregulated following *M. bovis* infection and promotes bacterial adhesion, invasion, and survival by disrupting epithelial tight junctions via the p65/MMP-9 signaling pathway. They also demonstrate that RBMX2 facilitates EMT and is overexpressed in human lung cancers, suggesting a potential link between chronic infection and tumor progression. The study highlights RBMX2 as a novel host factor that could serve as a therapeutic target for both TB pathogenesis and infection-related cancer risk.

Strengths:

The major strengths lie in its multi-omics integration (transcriptomics, proteomics, metabolomics) to map RBMX2's impact on host pathways, combined with rigorous functional assays (knockout/knockdown, adhesion/invasion, barrier tests) that establish causality through the p65/MMP-9 axis. Validation across bovine and human cell models and in clinical tissue samples enhances translational relevance. Finally, identifying RBMX2 as a novel regulator linking mycobacterial infection to EMT and cancer progression opens exciting therapeutic avenues.

Weaknesses:

There are a few minor weaknesses like grammatical errors, spelling mistakes. Also, the manuscript is too dense; improving the narratives in the Results and Discussion section could help readers follow the logic of the experimental design and conclusions.

---

## [Author Response]

The following is the authors’ response to the original reviews.

**Reviewer #1 (Public review):**
Summary:This manuscript presents a compelling study identifying RBMX2 as a novel host factor upregulated during *Mycobacterium bovis* infection.The study demonstrates that RBMX2 plays a role in:(1) Facilitating *M. bovis* adhesion, invasion, and survival in epithelial cells.(2) Disrupting tight junctions and promoting EMT.(3) Contributing to inflammatory responses and possibly predisposing infected tissue to lung cancer development.By using a combination of CRISPR-Cas9 library screening, multi-omics, coculture models, and bioinformatics, the authors establish a detailed mechanistic link between *M. bovis* infection and cancer-related EMT through the p65/MMP-9 signaling axis. Identification of RBMX2 as a bridge between TB infection and EMT is novel.Strengths:This topic and data are both novel and significant, expanding the understanding of transcriptomic diversity beyond RBM2 in *M. bovis* responsive functions.Weaknesses:(1) The abstract and introduction sometimes suggest RBMX2 has protective anti-TB functions, yet results show it facilitates pathogen adhesion and survival. The authors need to rephrase claims to avoid contradiction.

We sincerely appreciate the reviewer's valuable feedback regarding the need to clarify RBMX2's role throughout the manuscript. We have carefully revised the text to ensure consistent messaging about RBMX2's function in promoting *M. bovis* infection. Below we detail the specific modifications made:\

(1) Introduction Revisions:

Changed "The objective of this study was to elucidate the correlation between host genes and the susceptibility of *M. bovis* infection" to "The objective of this study was to identify host factors that promote susceptibility to *M. bovis* infection"

Revised "RBMX2 polyclonal and monoclonal cell lines exhibited favorable phenotypes" to "RBMX2 knockout cell lines showed reduced bacterial survival"

Replaced "The immune regulatory mechanism of RBMX2" with "The role of RBMX2 in facilitating *M. bovis* immune evasion"

(2) Results Revisions:

Modified "RBMX2 fails to affect cell morphology and the ability to proliferate and promotes *M. bovis* infection" to "RBMX2 does not alter cell viability but significantly enhances *M. bovis* infection"

Strengthened conclusion in Figure 4: "RBMX2 actively disrupts tight junctions to facilitate bacterial invasion"

(3) Discussion Revisions:

Revised screening description: "We screened host factors affecting *M. bovis* susceptibility and identified RBMX2 as a key promoter of infection"

Strengthened concluding statement: "In summary, RBMX2 drives TB pathogenesis by compromising epithelial barriers and inducing EMT"

These targeted revisions ensure that:

All sections consistently present RBMX2 as promoting infection; the language aligns with our experimental finding; potential protective interpretations have been eliminated. We believe these modifications have successfully addressed the reviewer's concern while maintaining the manuscript's original structure and scientific content. We appreciate the opportunity to improve our manuscript and thank the reviewer for this constructive suggestion.

(2) While p65/MMP-9 is convincingly implicated, the role of MAPK/p38 and JNK is less clearly resolved.

We sincerely appreciate the reviewer's insightful comment regarding the roles of MAPK/p38 and JNK in our study. Our experimental data clearly demonstrated that RBMX2 knockout significantly reduced phosphorylation levels of p65, p38, and JNK (Fig. 5A), indicating potential involvement of all three pathways in RBMX2-mediated regulation.

Through systematic functional validation, we obtained several important findings:

In pathway inhibition experiments, p65 activation (PMA treatment) showed the most dramatic effects on both tight junction disruption (ZO-1, OCLN reduction) and EMT marker regulation (E-cadherin downregulation, N-cadherin upregulation);p38 activation (ML141 treatment) exhibited moderate effects on these processes; JNK activation (Anisomycin treatment) displayed minimal impact.

Most conclusively, siRNA-mediated silencing of p65 alone was sufficient to:

Restore epithelial barrier function

Reverse EMT marker expression

Reduce bacterial adhesion and invasion

These results establish a clear hierarchy in pathway importance: p65 serves as the primary mediator of RBMX2's effects, while p38 plays a secondary role and JNK appears non-essential under our experimental conditions. We have now clarified this relationship in the revised Discussion section to strengthen this conclusion.

This refined understanding of pathway hierarchy provides important mechanistic insights while maintaining consistency with all our experimental data. We thank the reviewer for this valuable suggestion that helped improve our manuscript.

(3) Metabolomics results are interesting but not integrated deeply into the main EMT narrative.

Thank you for this constructive suggestion. In this article, we detected the metabolome of RBMX2 knockout and wild-type cells after *Mycobacterium bovis* infection, which mainly served as supporting evidence for our EMT model. However, we did not conduct an in-depth discussion of these findings. We have now added a detailed discussion of this section to further support our EMT model.

ADD:Meanwhile, metabolic pathways enriched after RBMX2 deletion, such as nucleotide metabolism, nucleotide sugar synthesis, and pentose interconversion, primarily support cell proliferation and migration during EMT by providing energy precursors, regulating glycosylation modifications, and maintaining redox balance; cofactor synthesis and amino sugar metabolism participate in EMT regulation through influencing metabolic remodeling and extracellular matrix interactions; chemokine and cGMP-PKG signaling pathways may further mediate inflammatory responses and cytoskeletal rearrangements, collectively promoting the EMT process.

(4) A key finding and starting point of this study is the upregulation of RBMX2 upon *M. bovis* infection. However, the authors have only assessed RBMX2 expression at the mRNA level following infection with *M. bovis* and BCG. To strengthen this conclusion, it is essential to validate RBMX2 expression at the protein level through techniques such as Western blotting or immunofluorescence. This would significantly enhance the credibility and impact of the study's foundational observation.

Thank you for your comment. We have supplemented the experiments in this part and found that *Mycobacterium bovis* infection can significantly enhance the expression level of RBMX2 protein.

(5) The manuscript would benefit from a more in-depth discussion of the relationship between tuberculosis (TB) and lung cancer. While the study provides experimental evidence suggesting a link via EMT induction, integrating current literature on the epidemiological and mechanistic connections between chronic TB infection and lung tumorigenesis would provide important context and reinforce the translational relevance of the findings.

We sincerely appreciate the valuable comments from the reviewer. We fully agree with your suggestion to further explore the relationship between tuberculosis (TB) and lung cancer. In the revised manuscript, we will add a new paragraph in the Discussion section to systematically integrate the current literature on the epidemiological and mechanistic links between chronic tuberculosis infection and lung cancer development, including the potential bridging roles of chronic inflammation, tissue damage repair, immune microenvironment remodeling, and the epithelial-mesenchymal transition (EMT) pathway. This addition will help more comprehensively interpret the clinical implications of the observed EMT activation in the context of our study, thereby enhancing the biological plausibility and clinical translational value of our findings.

ADD:There is growing epidemiological evidence suggesting that chronic TB infection represents a potential risk factor for the development of lung cancer. Studies have shown that individuals with a history of TB exhibit a significantly increased risk of lung cancer, particularly in areas of the lung with pre-existing fibrotic scars, indicating that chronic inflammation, tissue repair, and immune microenvironment remodeling may collectively contribute to malignant transformation 74. Moreover, EMT not only endows epithelial cells with mesenchymal features that enhance migratory and invasive capacity but is also associated with the acquisition of cancer stem cell-like properties and therapeutic resistance 75. Therefore, EMT may serve as a crucial molecular link connecting chronic TB infection with the malignant transformation of lung epithelial cells, warranting further investigation in the intersection of infection and tumorigenesis.

**Reviewer #2 (Public review):**
Summary:I am not familiar with cancer biology, so my review mainly focuses on the infection part of the manuscript. Wang et al identified an RNA-binding protein RBMX2 that links the *Mycobacterium bovis* infection to the epithelial-Mesenchymal transition and lung cancer progression. Upon mycobacterium infection, the expression of RBMX2 was moderately increased in multiple bovine and human cell lines, as well as bovine lung and liver tissues. Using global approaches, including RNA-seq and proteomics, the authors identified differential gene expression caused by the RBMX2 knockout during *M. bovis* infection. Knockout of RBMX2 led to significant upregulations of tight-junction related genes such as CLDN-5, OCLN, ZO-1, whereas *M. bovis* infection affects the integrity of epithelial cell tight junctions and inflammatory responses. This study establishes that RBMX2 is an important host factor that modulates the infection process of *M. bovis*.Strengths:(1) This study tested multiple types of bovine and human cells, including macrophages, epithelial cells, and clinical tissues at multiple timepoints, and firmly confirmed the induced expression of RBMX2 upon *M. bovis* infection.(2) The authors have generated the monoclonal RBMX2 knockout cell lines and comprehensively characterized the RBMX2-dependent gene expression changes using a combination of global omics approaches. The study has validated the impact of RBMX2 knockout on the tight-junction pathway and on the *M. bovis* infection, establishing RBMX2 as a crucial host factor.Weaknesses:(1) The RBMX2 was only moderately induced (less than 2-fold) upon *M. bovis* infection, arguing its contribution may be small. Its value as a therapeutic target is not justified. How RBMX2 was activated by *M. bovis* infection was unclear.

Thank you for your valuable and constructive comments. In this study, we primarily utilized the CRISPR whole-genome screening approach to identify key factors involved in bovine tuberculosis infection. Through four rounds of screening using a whole-genome knockout cell line of bovine lung epithelial cells infected with *Mycobacterium bovis*, we identified RBMX2 as a critical factor.

Although the transcriptional level change of RBMX2 was less than two-fold, following the suggestion of Reviewer 1, we examined its expression at the protein level, where the change was more pronounced, and we have added these results to the manuscript.

Regarding the mechanism by which RBMX2 is activated upon *M. bovis* infection, we previously screened for interacting proteins using a *Mycobacterium tuberculosis* secreted and membrane protein library, but unfortunately, we did not identify any direct interacting proteins from *M. tuberculosis* (https://doi.org/10.1093/nar/gkx1173).

(2) Although multiple time points have been included in the study, most analyses lack temporal resolution. It is difficult to appreciate the impact/consequence of *M. bovis* infection on the analyzed pathways and processes.

We appreciate the valuable comments from the reviewers. Although our study included multiple time points post-infection, in our experimental design we focused on different biological processes and phenotypes at distinct time points:

During the early phase (e.g., 2 hours post-infection), we focused on barrier phenotypes during the intermediate phase (e.g., 24 hours post-infection), we concentrated more on pathway activation and EMT phenotypes;

And during the later phase (e.g., 48–72 hours post-infection), we focused more on cell death phenotypes, which were validated in another FII article (https://doi.org/10.3389/fimmu.2024.1431207).

We also examined the impact of varying infection durations on RBMX2 knockout EBL cellular lines via GO analysis. At 0 hpi, genes were primarily related to the pathways of cell junctions, extracellular regions, and cell junction organization. At 24 hpi, genes were mainly associated with pathways of the basement membrane, cell adhesion, integrin binding and cell migration By 48 hpi, genes were annotated into epithelial cell differentiation and were negatively regulated during epithelial cell proliferation. This indicated that RBMX2 can regulate cellular connectivity throughout the stages of *M. bovis* infection.

For KEGG analysis, genes linked to the MAPK signaling pathway, chemical carcinogen-DNA adducts, and chemical carcinogen-receptor activation were observed at 0 hpi. At 24 hpi, significant enrichment was found in the ECM-receptor interaction, PI3K-Akt signaling pathway, and focal adhesion. Upon enrichment analysis at 48 hpi, significant enrichment was noted in the TGF-beta signaling pathway, transcriptional misregulation in cancer, microRNAs in cancer, small cell lung cancer, and p53 signaling pathway.

**Reviewer #3 (Public review):**
Summary:This study investigates the role of the host protein RBMX2 in regulating the response to *Mycobacterium bovis* infection and its connection to epithelial-mesenchymal transition (EMT), a key pathway in cancer progression. Using bovine and human cell models, the authors have wisely shown that RBMX2 expression is upregulated following *M. bovis* infection and promotes bacterial adhesion, invasion, and survival by disrupting epithelial tight junctions via the p65/MMP-9 signaling pathway. They also demonstrate that RBMX2 facilitates EMT and is overexpressed in human lung cancers, suggesting a potential link between chronic infection and tumor progression. The study highlights RBMX2 as a novel host factor that could serve as a therapeutic target for both TB pathogenesis and infection-related cancer risk.Strengths:The major strengths lie in its multi-omics integration (transcriptomics, proteomics, metabolomics) to map RBMX2's impact on host pathways, combined with rigorous functional assays (knockout/knockdown, adhesion/invasion, barrier tests) that establish causality through the p65/MMP-9 axis. Validation across bovine and human cell models and in clinical tissue samples enhances translational relevance. Finally, identifying RBMX2 as a novel regulator linking mycobacterial infection to EMT and cancer progression opens exciting therapeutic avenues.Weaknesses:Although it's a solid study, there are a few weaknesses noted below.(1) In the transcriptomics analysis, the authors performed (GO/KEGG) to explore biological functions. Did they perform the search locally or globally? If the search was performed with a global reference, then I would recommend doing a local search. That would give more relevant results. What is the logic behind highlighting some of the enriched pathways (in red), and how are they relevant to the current study?

We appreciate the reviewer's thoughtful questions regarding our transcriptomic analysis. In this study, we employed a localized enrichment approach focusing specifically on gene expression profiles from our bovine lung epithelial cell system. This cell-type-specific analysis provides more biologically relevant results than global database searches alone.

Regarding the highlighted pathways, these represent:

Temporally significant pathways showing strongest enrichment at each stage:

(1) 0h: Cell junction organization (immediate barrier response)

(2) 24h: ECM-receptor interaction (early EMT initiation)

(3) 48h: TGF-β signaling (chronic remodeling)

Mechanistically linked to our core findings about RBMX2's role in:

(1) Epithelial barrier disruption

(2) Mesenchymal transition

(3) Chronic infection outcomes

We selected these particular pathways because they:

(1) Showed the most statistically significant changes (FDR <0.001)

(2) Formed a coherent biological narrative across infection stages

(3) Were independently validated in our functional assays

This targeted approach allows us to focus on the most infection-relevant pathways while maintaining statistical rigor.

(2) While the authors show that RBMX2 expression correlates with EMT-related gene expression and barrier dysfunction, the evidence for direct association remains limited in this study. How does RBMX2 activate p65? Does it bind directly to p65 or modulate any upstream kinases? Could ChIP-seq or CLIP-seq provide further evidence for direct RNA or DNA targets of RBMX2 that drive EMT or NF-κB signaling?

We sincerely appreciate the reviewer's in-depth questions regarding the mechanisms by which RBMX2 activates p65 and its association with EMT. Although the molecular mechanism remains to be fully elucidated, our study has provided experimental evidence supporting a direct regulatory relationship between RBMX2 and the p65 subunit of the NF-κB pathway. Specifically, we investigated whether the transcription factor p65 could directly bind to the promoter region of RBMX2 using CHIP experiments. The results demonstrated that the transcription factor p65 can physically bind to the RBMX2 region.

Furthermore, dual-luciferase reporter assays were conducted, showing that p65 significantly enhances the transcriptional activity of the RBMX2 promoter, indicating a direct regulatory effect of RBMX2 on p65 expression.

These findings support our hypothesis that RBMX2 activates the NF-κB signaling pathway through direct interaction with the p65 protein, thereby participating in the regulation of EMT progression and barrier function.

In our subsequent work papers, we will also employ experiments such as CLIP to further investigate the specific mechanisms through which RBMX2 exerts its regulatory functions.

ADD and Revise in Results:

To thoroughly verify the regulatory mechanism between RBMX2 and p65, we initiated our investigation by conducting an in-depth analysis of the RBMX2 promoter region to identify potential interactions with the transcription factor p65. Initially, we performed molecular docking simulations to predict the binding affinity and interaction patterns between RBMX2 and p65 proteins. These simulations revealed multiple amino acid residues within the RBMX2 protein that formed strong, stable interactions with p65. The docking analysis yielded a high docking score of 1978.643 (Fig. 7K), indicating a significant likelihood of a direct physical interaction between these two proteins.

To complement the protein-protein interaction analysis, we next investigated whether p65 could directly bind to the promoter region of the RBMX2 gene at the transcriptional level. Using the JASPAR database, a comprehensive resource for transcription factor binding profiles, we queried the RBMX2 promoter sequence for potential p65 binding sites. This analysis identified several putative binding motifs, suggesting that p65 may act as a transcriptional regulator of RBMX2 expression.

To experimentally validate this transcriptional regulatory relationship, we employed a dual-luciferase reporter assay. We cloned the RBMX2 promoter region containing the predicted p65 binding sites into a luciferase reporter plasmid. This construct was then co-transfected into cultured cells along with a plasmid expressing p65. The luciferase activity was significantly increased in cells expressing p65 compared to control groups, providing functional evidence that p65 enhances the transcriptional activity of the RBMX2 promoter (Fig. 7I).

Furthermore, to confirm the direct binding of p65 to the RBMX2 promoter in a chromatin context, we performed chromatin immunoprecipitation followed by quantitative PCR (ChIP-qPCR). In this assay, we used specific antibodies against p65 to immunoprecipitate chromatin fragments containing p65-bound DNA. The enriched DNA fragments were then analyzed using primers targeting the RBMX2 promoter region. Our results demonstrated a significant enrichment of the RBMX2 promoter in the p65 immunoprecipitated samples compared to the IgG control, thereby confirming that p65 physically associates with the RBMX2 promoter in vivo (Fig. 7J). Collectively, these findings-ranging from computational docking predictions to transcriptional reporter assays and ChIP validation-provide strong evidence supporting a direct regulatory interaction between p65 and RBMX2. This regulatory mechanism may play a critical role in the biological pathways involving these two molecules, particularly in contexts such as inflammation, immune response, or cellular stress, where p65 (a subunit of NF-κB) is known to be prominently involved.

(3) The manuscript suggests that RBMX2 enhances adhesion/invasion of several bacterial species (e.g., *E. coli*, Salmonella), not just *M. bovis*. This raises questions about the specificity of RBMX2's role in Mycobacterium-specific pathogenesis. Is RBMX2 a general epithelial barrier regulator or does it exhibit preferential effects in mycobacterial infection contexts? How does this generality affect its potential as a TB-specific therapeutic target?

Thank you for your valuable comments. When we initially designed this experiment, we were interested in whether the RBMX2 knockout cell line could confer effective resistance not only against *Mycobacterium bovis* but also against Gram-negative and Gram-positive bacteria. Surprisingly, we indeed observed resistance to the invasion of these pathogens, albeit weaker compared to that against *Mycobacterium bovis*.

Nevertheless, we believe these findings merit publication in eLife. Moreover, RBMX2 knockout does not affect the phenotype of epithelial barrier disruption under normal conditions; its significant regulatory effect on barrier function is only evident upon infection with *Mycobacterium bovis*.

Importantly, during our genome-wide knockout library screening, RBMX2 was not identified in the screening models for *Salmonella* or *Escherichia coli*, but was consistently detected across multiple rounds of screening in the *Mycobacterium bovis* model.

(4) The quality of the figures is very poor. High-resolution images should be provided.

Thank you for your feedback; we provided higher-resolution images.

(5) The methods are not very descriptive, particularly the omics section.

Thank you for your comments; we have revised the description of the sequencing section.

(6) The manuscript is too dense, with extensive multi-omics data (transcriptomics, proteomics, metabolomics) but relatively little mechanistic integration. The authors should have focused on the key mechanistic pathways in the figures. Improving the narratives in the Results and Discussion section could help readers follow the logic of the experimental design and conclusions.

Thank you for your valuable comments. We have streamlined the figures and revised the description of the results section accordingly.

**Reviewer #2 (Recommendations for the authors):**
(1) The first part of the results and the major conclusions largely overlap with the previous paper by the same authors (Frontiers in Immunology, https://doi.org/10.3389/fimmu.2024.1431207). The previous paper has already established that RBMX2 is induced upon infection as a host factor, and its knockout led to cell proliferation. Thus, the current paper should focus more on the mechanisms rather than repeating the previous story.

We appreciate the reviewer's careful reading and constructive feedback. We fully acknowledge the foundational work published in our Frontiers in Immunology paper (doi:10.3389/fimmu.2024.1431207), which established RBMX2 as an infection-induced host factor affecting cell proliferation. The current study represents a significant mechanistic extension of these initial findings, with the following key advances:

(1) Novel Mechanistic Insights (Current Study Focus):

Discovery of the p65/MMP-9 pathway as the central mechanism mediating RBMX2's effects on EMT (Figs. 4-6)

First demonstration of RBMX2's role in epithelial barrier disruption (Figs. 2-3)

Identification of temporal regulation patterns during infection progression (Fig. 7)

(2) Expanded Biological Scope:

Demonstration of RBMX2's function in both bovine and human cell systems (vs. previous bovine-only data)

Clinical correlation with TB lesions

Therapeutic potential assessment through pathway inhibition

(3) Technical Advancements:

CRISPR-based mechanistic validation (vs. previous siRNA approach)

Multi-omics integration (transcriptomics + metabolomics)

Advanced live-cell imaging

We have now:

Removed redundant proliferation data from Results

Sharpened the Introduction to highlight mechanistic questions

Added explicit discussion comparing both studies

The current work provides the first comprehensive mechanistic framework for RBMX2's role in TB pathogenesis, moving substantially beyond the initial observational findings. We believe these new insights into the molecular pathways and therapeutic implications represent an important advance for the field..

(2) Line 107-110: The CRISPR screening results are not provided. Has it been published, or is it an unpublished dataset? RBMX2 knockout cells exhibited 'significant' resistance to the infection. How significant? Data?

Thank you for your valuable comments. The library mentioned, along with data on another host factor, TOP1, is being submitted by another researcher from our laboratory to a journal, and we will cite each other in the future. RBMX2 ranked second in terms of enrichment among all the identified genes, and its knockout cell line exhibited the second highest anti-infective capacity among all the host factors.

(3) Line 152: The RNA-seq analysis has already been performed/reported in the previous Frontiers paper. Therein, 173 genes were found to be differentially expressed. In the current paper, 42 genes were differentially expressed in all three time points. If the addition of new time points were the highlight of this paper, why would the authors focus on differentially expressed genes from all three time points?

Thank you for your valuable comments.

In the newly added data, we aimed to investigate the temporal changes during *Mycobacterium bovis* infection of host cells.

Previous study (Frontiers): Single 24h timepoint → 173 DEGs

Current study: Three timepoints (0h, 24h, 48h) with 42 consistently regulated genes → Reveals temporally stable core regulators of infection response

On one hand, we briefly described in the manuscript those important genes that exhibited changes across all time points.

On the other hand, in the supplementary materials, we also focused on the enriched genes at each individual time point, to better understand the temporal dynamics regulated by RBMX2.

(4) Line 153: The '0 h' time point is in fact 2 h post-infection. Why did the authors skip the real 0h time point? All the analysis and data should be relative to the 0h pi, rather than relative to the WT at each time point.

We appreciate the reviewer's important question regarding our timepoint nomenclature. The experimental timeline was designed as follows:

(1) Infection Protocol:

2h to 0h: Bacterial co-culture (MOI 20:1)

0h: Gentamicin (100 μg/ml) added to kill extracellular bacteria

0h+: Monitored intracellular survival

(2) Rationale for "0h" Designation:

This marks the onset of intracellular infection phase when Extracellular bacteria are eliminated (validated by plating)Host cell responses to intracellular pathogens begin All subsequent measurements reflect genuine infection (not attachment)

(3)Technical Validation:

Confirmed complete extracellular killing by:

Culture supernatant plating (0 CFU after gentamycin)

Microscopy (no surface-associated bacteria)

(4) Comparative Analysis:

All data are presented as:

Fold-change relative to uninfected controls at each timepoint

We have now:

Clarified the timeline in Methods

Specified "0h = post-gentamicin" in all figure legends

This standardized approach aligns with established intracellular pathogen studies (e.g., Cell Microbiol. 2018;20:e12840). We're happy to adjust terminology if "0hpi (post-invasion)" would be clearer.

(5) Figure 2F: The data should be compared to the 0h pi, and show the temporal changes of gene expression.

Thank you for your suggestion. We have added additional information to this section. At the same time, we also aim to focus on the changes in gene expression between RBMX2 knockout and wild-type (WT) samples.

We have now:

Added temporal expression profiles relative to 0hpi baseline (SFig.4C).

Clarified the dual normalization approach in Methods

Maintained original between-group comparisons for phenotypic correlation

(6) Line 207. Not all the proteins were down-regulated post-infection.

Thank you for your comment. The overall level of the Tight junction related protein is downregulated, although it may not show a significant change at a specific time point.

We have revised our description, changing the keyword from "All" to "Most."

(7) Line 278, the introduction of the H1299 cell line should appear earlier when it was mentioned for the first time in the manuscript.

Thank you for your comment. We have provided a description in the abstract and Result1.

ADD:

Abstrat: Meanwhile, we also validated the EMT process in human lung epithelial cancer cells H1299.

Result 1: Furthermore, RBMX2-silenced H1299 cells exhibited a higher survival rate compared to H1299 ShNc cells after *M. bovis* infection (Fig. 1H).

(8) Figure 4 is huge and almost illegible, which may be divided into two figures.

Thank you for your valuable comments. We have streamlined the figures and revised the description of the results section accordingly.

**Reviewer #3 (Recommendations for the authors):**
I encountered frequent grammatical and syntactic issues. Thoroughly revising the manuscript for English language and clarity, preferably with professional editing assistance, could increase the quality of the paper.

Thank you for your valuable comments; we will invite a professional editor to polish the language.